# Polymer model integrates imaging and sequencing to reveal how nanoscale heterochromatin domains influence gene expression

Vinayak Vinayak [1,2], Ramin Basir [1,2], Rosela Golloshi[3,4], Joshua Toth [1,2], Lucas Sant'Anna [1,5], Melike Lakadamyali [1,6], Rachel Patton McCord [7] & Vivek B. Shenoy [1,2] ✉

Chromatin organization regulates gene expression, with nanoscale heterochromatin domains playing a fundamental role. Their size varies with microenvironmental stiffness and epigenetic interventions, but how these factors regulate their formation and influence transcription remains unclear. To address this, we developed a sequencing-informed copolymer model that simulates chromatin evolution through diffusion and active epigenetic reactions. Our model predicts the formation of nanoscale heterochromatin domains and quantifies how domain size scales with epigenetic reaction rates, showing that epigenetic and compaction changes primarily occur at domain boundaries. We validated these predictions via Hi-C and super-resolution imaging of hyperacetylated melanoma cells and identified differential expression of metastasis-related genes through RNA-seq. We validated our findings in hMSCs, where epigenetic reaction rates respond to microenvironmental stiffness. Conclusively, our simulations reveal that heterochromatin domain boundaries regulate gene expression and epigenetic memory. These findings demonstrate how external cues drive chromatin organization and transcriptional memory in development and disease.

The three-dimensional organization of the mammalian genome within the nucleus is a critical determinant of cell fate, regulating transcription and thereby influencing development, differentiation, metabolism, and proliferation. The segregation of active and repressed genes into distinct phases, comprising loosely packed, transcriptionally active, euchromatin (A-compartment) and compact, transcriptionally repressed, heterochromatin (B-compartment) phases, has been established by complimentary approaches[1,2], including next-generation sequencing techniques (Hi-C[3–6], Micro-C[7], ATAC-seq[8]) and super-resolution imaging[9–11]. Epigenetic profiling experiments such as ChIP-seq[12,13] and immunofluorescence imaging have established that phase-separated regions exhibit unique epigenetic signatures[14,15]. Specifically, heterochromatin is characterized by persistent histone methylation (such as H3K9me3 or H3K27me3), whereas euchromatin is characterized by prominent histone acetylation (such as H3K27ac)[16–19]. These marks are instrumental in governing

[1]Center for Engineering Mechanobiology, University of Pennsylvania, Philadelphia, PA, USA. [2]Department of Materials Science and Engineering, University of Pennsylvania, Philadelphia, PA, USA. [3]Departments of Cell Biology, Center for Cell Dynamics, Johns Hopkins University School of Medicine, Baltimore, MD, USA. [4]Giovanis Institute for Translational Cell Biology, Johns Hopkins Medicine, Baltimore, MD, USA. [5]Department of Bioengineering, Stanford University, Stanford, CA, USA. [6]Department of Physiology, Perelman School of Medicine, University of Pennsylvania, Philadelphia, PA, USA. [7]Department of Biochemistry & Cellular and Molecular Biology, University of Tennessee, Knoxville, TN, USA. ✉e-mail: vshenoy@seas.upenn.edu

transcriptional likelihood[20]. Super-resolution imaging has revealed the widespread presence of heterochromatin-rich packing domains as fundamental units of chromatin organization[21–23]. These domains have been observed across diverse cell lines, although this aspect of chromatin organization remains poorly understood.

Chromatin organization exhibits dynamic responsiveness to a broad spectrum of physical and chemical extracellular cues, orchestrating a complex interplay between epigenetic remodeling and transcriptional regulation. Recent studies have elucidated the pivotal role of mechanical stimuli, including substrate elasticity[23] and viscosity[24], in directing the redistribution of histone epigenetic marks via the action of histone epigenetic remodelers such as histone deacetylases (HDACs) and histone methyltransferases (HMTs) (e.g., HDAC3 and EZH2), which are also known as "histone writers" and "histone erasers"[23–27]. Furthermore, diseased states such as fatty liver and cancer metastasis, characterized by altered extracellular lipid concentrations[28] and intra/extravasation[29], also exhibit large-scale chromatin reorganization. Notably, pharmacological, and epigenetic interventions evoke similar genomic and epigenomic changes[23], underscoring the central role of epigenetic remodelers in modulating histone marks through methylation and acetylation. These epigenetic changes, in turn, reconfigure nanoscale chromatin domains[30–33], influencing transcriptional outcomes and cell fate. Despite significant advances, the physical mechanisms underlying this multiscale process remain poorly understood, warranting further investigation to elucidate the intricate relationships among chromatin dynamics, epigenetic regulation, and disease pathogenesis.

Modeling efforts in recent years have largely viewed chromatin organization as a passive, equilibrium system, yielding valuable insights into the interplay between epigenetic marks and 3D chromatin structure[34–40]. However, a smaller subset of studies has explored chromatin organization in a more realistic setting, incorporating the active energy flux driven by epigenetic remodelers, which better captures the dynamic nature of chromatin organization. Computational models have begun to explore such active dynamics[41–44], including the impact of cohesin-mediated chromatin looping[45–49] and transcriptional activity[50–52] on chromatin organization, including the phenomenon of epigenetic spreading[44,53,54]. However, there is a significant gap in the understanding of how epigenetic reactions, which are influenced by changes in the microenvironment, drug exposure, disease progression, and aging, regulate 4D chromatin organization. This omission overlooks the crucial role of histone epigenetic remodelers in shaping chromatin organization through the formation of heterochromatin-rich packing domains and the role of such an organization in controlling the cell's transcriptional activity, underscoring the need for a more comprehensive biophysical approach to capture this complex interplay.

Here, we present a chromatin copolymer model that integrates bioinformatics inputs with epigenetic reaction-driven dynamics to predict spatio-temporal genomic organization. The polymer comprising euchromatin and heterochromatin segments is informed by experimentally obtained Hi-C or ChIP-seq data, ensuring a biologically realistic representation and facilitating direct comparison with experimental observations. Our model captures polymer energetics through chromatin-chromatin interactions, and passive dynamics through a combination of nucleoplasmic and epigenetic diffusion, driving phase separation. Epigenetic reactions capture the effect of histone epigenetic remodelers, actively interconverting euchromatin and heterochromatin. This reaction-driven approach enables us to capture the effects of both mechanical and drug-induced changes on chromatin organization. We demonstrate that a delicate balance between the diffusion of epigenetic marks and epigenetic reactions gives rise to characteristic chromatin domains and establishes a scaling relationship between their size distribution and reaction rates. We validate these findings through super-resolution (STORM) imaging of

chromatin domains in human A375 malignant melanoma cells after hyperacetylation through an epigenetic drug, which reveals the global decompaction of chromatin domains. Bulk Hi-C sequencing of these cells shows excellent agreement with our model predictions, capturing epigenetic compartmental shifts with high accuracy at a kilobase-scale resolution. Notably, our model predicts that epigenetic changes primarily occur at heterochromatin domain boundaries, which is validated by Hi-C contact maps. Further RNA-seq analysis revealed a correlation between chromatin decompaction and the differential regulation of EMT-related genes in melanoma, near domain boundaries. To validate our model's predictive power in diverse microenvironments, we simulated changes in chromatin organization in human mesenchymal stem cells in response to altered substrate stiffness and experimentally confirm these predictions via super-resolution imaging. Finally, our chromatin polymer model reveals the role of chromatin organization in the formation of history-dependent epigenetic memory, providing a general framework for understanding how extracellular cues shape temporal chromatin organization and gene expression.

## Results

### 2a Chromatin polymer model follows experimentally observed length scales and incorporates diffusion and epigenetic reaction dynamics

Chromatin exhibits complex organization across multiple scales, as revealed through conformation capture maps such as Hi-C and Micro-C[55]. In particular, the A (euchromatin-like and active) and B (heterochromatin-like and repressed) compartments emerge as prominent structures characterized by enhanced chromatin interactions within their respective types. These interactions are physically facilitated by bridging proteins such as HP1[56] and PRC1[57] in addition to other physical interactions, which lead to large-scale phase separation[58,59]. Simultaneously, experimental investigations such as histone ChIP-seq[60] have demonstrated a significant overlap between active histone tail marks (e.g., H3K27ac) and the A compartment, whereas repressive histone tail marks (e.g., H3K9me3 or H3K27me3) tend to associate with the B compartments[3].

Based on these findings and other computational studies[35,41], we choose to represent chromatin as a block copolymer, with active/A and repressed/B regions. Specifically, we represent chromatin as a self-avoiding beads-on-a-string polymer, where the beads can be labeled as active/euchromatin-like (blue) or repressed/heterochromatin-like (red) (Fig. 1a). Each bead in the simulation corresponds to a continuous genomic region of 10 kilobase pairs (kb), with a bead size (σ) set at 65 nm. More details on the polymer configuration are provided in Supplementary Information 1.1-1.2. To explore genome organization principles, we first modeled a randomly initialized chromatin polymer (3000 beads, 60% heterochromatin) to probe the biophysical mechanisms of chromatin packing. Next, we incorporated experimental data-driven initialization, assigning chromatin states based on ChromHMM annotations (for hMSCs) or Hi-C A/B compartmentalization (for A375 melanoma cells). These complementary approaches validate the robustness of the model. While more than two epigenetic flavors derived from different ChromHMM states[61,62], multiple histone ChIP-seq datasets or finer Hi-C or MicroC compartmentalization can be incorporated into the model, our focus remains on highlighting the key role of heterochromatin domains in regulating changes in gene expression; hence, we stick to a two-component model.

To quantify chromatin interactions, we implemented a Lennard-Jones potential function, capturing both bridging protein-mediated interactions and nonspecific chromatin-chromatin affinities. Excluded volume interactions were imposed between euchromatin and heterochromatin, with stronger interactions within heterochromatin ($\epsilon_{HH} > \epsilon_{EE}$), hence maintaining a higher packing density for heterochromatin[19,63,64] and driving phase separation (Fig. 1b, c). The

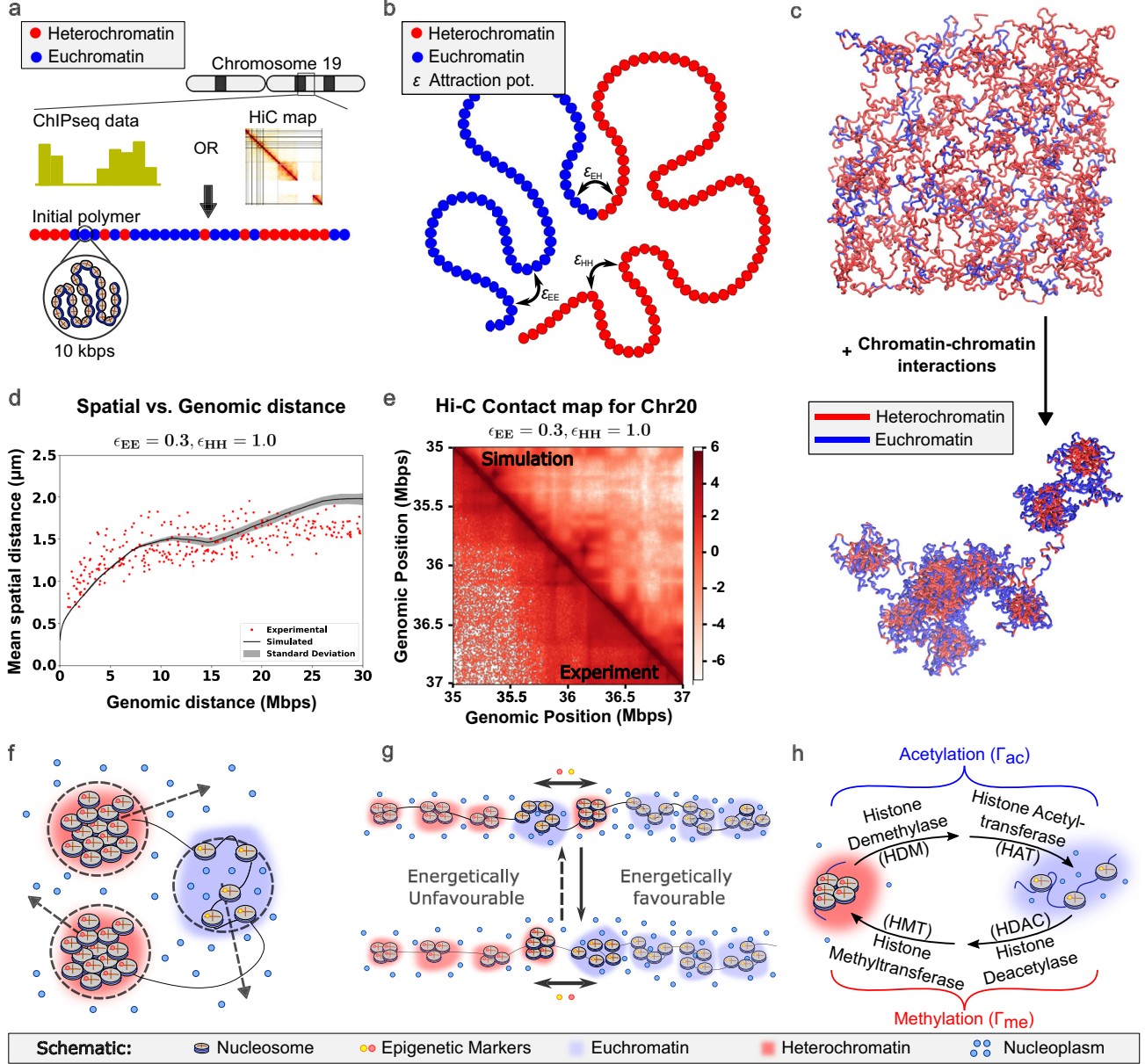

**Fig. 1 | Polymer model based on experimentally obtained inputs and biologically relevant dynamics. a** The initial polymer strand is constructed using inputs from experimentally obtained sequencing data. The histone ChIP-seq data (through ChromHMM[61]) or Hi-C data are used to label the initial configuration of the polymer. We assign all repressed states to heterochromatin (red) and active states to euchromatin (blue) beads. **b** Lennard–Jones pairwise potentials are defined for each pair of beads based on their epigenetic marking. $\epsilon_{EE}$, $\epsilon_{EH}$ and $\epsilon_{HH}$ are the euchromatin-euchromatin, euchromatin-heterochromatin and heterochromatin-heterochromatin bead interaction potential strengths, respectively. For all simulations in the manuscript, we choose $\epsilon_{EE} = 0.3$ and $\epsilon_{EH} = 1.0$. **c** The system is relaxed (bottom) from a random initial configuration (top) via Langevin dynamics. **d** Mean spatial distance vs. mean genomic distance obtained after

polymer relaxation is plotted with the experimental data[18]. Simulation data presented as mean;+/− SD across 10 runs. **e** The simulated Hi-C map (upper triangle) shows the characteristic checkerboard pattern, which is observed in the experimental Hi-C map (lower triangle)[3]. **f** Diffusion of the nucleoplasm (water) is accounted for implicitly via Brownian dynamics. **g** Diffusion of epigenetic marks is modeled via an energy-based metropolis criterion for exchanging epigenetic marks between spatially neighboring beads. **h** Epigenetic reactions of acetylation (through the action of histone demethylase (HDM) and histone acetyltransferase (HAT)) and methylation (through the action of histone deacetylase (HDAC) and histone methyltransferase (HMT)) are modeled as Monte Carlo-based epigenetic reassignment processes.

polymer was relaxed via Langevin dynamics, producing an equilibrium configuration that recapitulates experimentally observed spatial chromatin organization, including the genomic distance–spatial distance relationship (Fig. 1d) and the characteristic Hi-C checkerboard pattern of A/B compartmentalization (Fig. 1e). More details on the energetics of the model are provided in the methods section Data-driven construction of the polymer model.

Having defined the energetics of the chromatin segments, we next consider the dynamics of spatial and epigenetic reorganization

over time. The spatial displacement of chromatin within the nucleoplasm is governed by Brownian diffusion, where chromatin beads move within a viscous environment, subject to molecular crowding and chromatin-chromatin interactions (Fig. 1f). This diffusion facilitates the rearrangement of chromatin segments, promoting heterochromatin domain formation through preferential self-association of repressed regions. Concurrently, epigenetic marks redistribute across chromatin, pushing the system toward a lower energy configuration in which beads with similar epigenetic marks

coalesce. This process, which we term the diffusion of epigenetic marks, is captured via a Monte Carlo-based epigenetic remarking algorithm (Fig. 1g). In addition, epigenetic reactions introduce active remodeling, allowing euchromatin to be converted to heterochromatin and vice versa (Fig. 1h). These reactions alter the net heterochromatin-to-euchromatin ratio and are influenced by external factors such as the mechanical properties of the microenvironment[23,25,26,28] and the action of epigenetic drug treatments[22,23]. We model these dynamics by quantifying the conversion of heterochromatin to euchromatin through the acetylation rate ($\Gamma_{ac}$) and the conversion of euchromatin to heterochromatin through the methylation rate ($\Gamma_{me}$). These reaction terms effectively capture the activity of histone-modifying enzymes, including histone deacetylases (HDACs), histone acetyltransferases (HATs), histone methyltransferases (HMTs), and histone demethylases (HDMs) which regulate chromatin accessibility through post-translational modifications. The detailed computational implementation of these processes is described in the methods section Diffusion and Epigenetic Reactions.

## 2b Interplay of diffusion and reaction dynamics drives chromatin domain formation and size scaling

We start by quantifying the 3D spatial organization principles defining the nanoscale behavior of the chromatin polymer model through analysis of its temporal evolution. We illustrate how diffusion and reaction dynamics (as described in section Chromatin polymer model follows experimentally observed length scales and 119 incorporates diffusion and epigenetic reaction dynamics) work in tandem to create

nanoscale heterochromatic domains, resulting in a scaling relationship between domain size and the rates of diffusion and epigenetic reactions. To distinguish this physical phenomenon from those that depend on specific details of heterochromatin and euchromatin segment arrangements, we first investigate the prototype random polymer model (described in Chromatin polymer model follows experimentally observed length scales and 119 incorporates diffusion and epigenetic reaction dynamics and Supplementary Information 1.1). For the prototype polymer, we choose the heterochromatin-to-euchromatin ratio to be 1.5, mirroring the prevalent observation in most Hi-C maps (Supplementary Information 1.1).

**Diffusion of epigenetic marks leads to Ostwald ripening.** We start by considering only spatial (Brownian) diffusion of chromatin and the diffusion of epigenetic marks. This ensures that the system adheres to thermodynamic balance, and the global ratio of active (blue) and repressed (red) chromatin segments (beads) remains constant throughout the simulation. Commencing with the prototype polymer in a random configuration, as illustrated in the left panel of Fig. 2a, we track the temporal evolution by monitoring the radius of gyration of the heterochromatin-rich domains. The methodology for calculating the radius of gyration is described in Supplementary Information 1.5. As the system evolves, it reaches a steady-state configuration (Fig. 2a, red simulation trajectory), which remains constant over time. Upon closer examination of the trajectory, we observe initial clustering and enlargement of the repressed regions facilitated by the diffusion of epigenetic marks, as quantified by the initial growth in the radius of gyration of the repressed domains.

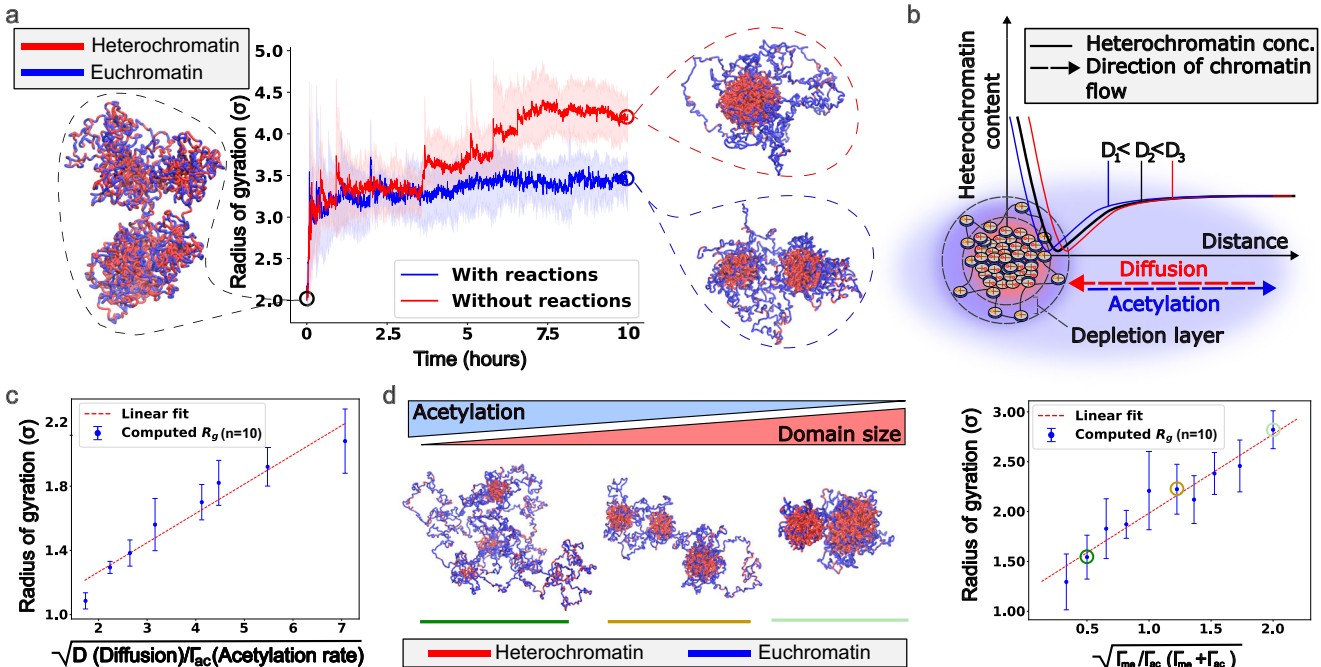

**Fig. 2 | Chromatin dynamics form finite-sized domains and exhibit domain scaling. a** The evolution of the radius of gyration of the domains of the prototype polymer in the presence and absence of epigenetic reactions. Starting from a randomly labeled (60% heterochromatin) polymer, the proposed dynamics lead to the ripening of the heterochromatin domain in the absence of reactions and readily form finite-sized domains in their presence (reaction ratio is kept constant with $\Gamma_{me}/\Gamma_{ac}=1.2$). The plots show mean +/− SD across 10 simulations. **b** Schematic showing the action of diffusion and acetylation on the heterochromatin packing domains. The y-axis shows the heterochromatin concentration, and the x-axis shows the distance from the center of the domain. Diffusion and acetylation act in opposite directions, with diffusion leading to domain growth and acetylation to

shrinkage. The empirical heterochromatin content vs. distance plots have been provided for three different diffusion rates (D) with the same reaction rates. **c** A scaling relationship is observed between the radius of gyration and the relative rates of diffusion and reactions, which dictates that the radii of the domains scale linearly with $\sqrt{D/\Gamma_{ac}}$. The data points represent mean +/− SD. **d** With constant diffusion, as the reaction rates change, the domain sizes change. Higher methylation (lower acetylation) drives an increase in domain size (polymer snapshots). A scaling relation is obtained, which shows that the average radius of gyration of the heterochromatin domains is directly proportional to $\sqrt{\Gamma_{me}/\Gamma_{ac}(\Gamma_{me}+\Gamma_{ac})}$ (right). The data points represent mean +/− SD. Circled points on the plot represent the parameters for the polymer configurations on the left.

Complete spatial segregation of the epigenetic marks subsequently occurs, with the repressed marks migrating toward the center of the simulation box while active marks are displaced outward, depicted through the plateauing of the radius of gyration. This spatial segregation represents a thermodynamically favored state of the system, which minimizes the interfacial area between the active and repressed phases, reminiscent of Ostwald ripening. These simulations demonstrate that passive diffusion alone fails to prevent the coarsening of epigenetic marks and, thus, does not lead to the formation of characteristic chromatin domains.

**Epigenetic reactions compete with diffusion to give rise to domains of a characteristic size.** Next, we introduce epigenetic reactions into our system. These reactions lead to out-of-equilibrium interconversion of epigenetic mark dynamics, which can alter the global ratio of repressed (red) and active (blue) polymer beads (details on detailed balance considerations in Supplementary Information1.4). Notably, in the previous scenario where no reactions were present, the net heterochromatin-to-euchromatin ratio remained the same as that of the initial polymer. However, with the introduction of reactions, this ratio becomes variable and stabilizes at a steady state equal to the ratio of the rates of the methylation and acetylation reactions. In this context, we maintain the reaction ratio identical to the initial polymer composition to facilitate an unbiased comparison with the diffusion-only scenario, i.e., $\Gamma_{me}/\Gamma_{ac} = 1.5$. The initial polymer configuration is also the same as that of the diffusion-only setting (Fig. 2a). Like in the diffusion-only case, we observe an initial ripening phase where the repressed marks diffuse to form phase-separated domains (Fig. 2a, blue simulation trajectory). However, as the simulation progresses, the growth of these domains levels off, resulting in distinct and persistent separated domains. These repressed domains exhibit a characteristic length scale akin to what is observed in super-resolution imaging studies[21–23]. Moreover, the simulation illustrates that these repressed domains maintain their characteristic shapes over an extended period (~ 10 h of real-time simulation), comparable to the timescale of a typical duration of human cell interphase. This suggests that by incorporating the out-of-equilibrium dynamics of epigenetic reactions alongside equilibrium-preserving diffusion dynamics, the simulation yields heterochromatin domains that are stable against Ostwald ripening.

To understand the biophysical underpinnings of domain formation in more detail, we next elucidate the underlying mechanisms. During the initial growth phase of the domains (Fig. 2a), favorable interactions facilitate the clustering of similar epigenetic marks through energetically driven Brownian diffusion and the diffusion of epigenetic marks. This aggregation leads to the formation of heterochromatin-rich domains surrounded by heterochromatin-poor regions (Fig. 2b). The depletion of heterochromatin from the domain's immediate vicinity establishes a flux of heterochromatin regions toward the formed domain, driving its growth. In the absence of reactions, this flux leads to complete phase separation. However, when reactions are introduced, the acetylation reaction (as quantified by $\Gamma_{ac}$), which converts heterochromatin to euchromatin in a spatially invariant manner, converts heterochromatin (red) to euchromatin (blue) within the growing domains, competing with the diffusion process. As the simulation progresses, the reaction kinetics hinder complete phase separation. Domain growth ceases when the influx of heterochromatin through diffusion is balanced by the conversion of heterochromatin to euchromatin within the domain via the acetylation reaction. A scaling argument (Supplementary Information1.6) shows that the radii of the heterochromatic domains $R_d$ follow the scaling relation[23]:

$$R_d \propto \sqrt{\frac{D}{\Gamma_{ac}} \frac{\Gamma_{me}}{(\Gamma_{me} + \Gamma_{ac})}} \qquad (1)$$

where $D$ is the diffusion rate of epigenetic marks and $\Gamma_{me}$ and $\Gamma_{ac}$ are the methylation and acetylation rates, respectively (as explained in the section Chromatin polymer model follows experimentally observed length scales and 119 incorporates diffusion and epigenetic reaction dynamics). To ascertain this diffusion–reaction relationship through our proposed MD model, we fix the reaction rates and vary only the diffusion rates in our simulation (Fig. 2c). The simulation confirms the nonlinear relationship of the radius, which increases with diffusion and is counteracted by acetylation, i.e., proportional to $\sqrt{D/\Gamma_{ac}}$. Here, we note that histone turnover rates (which have a wide range of reported values[65–67]), which influence reaction rates, play a crucial role in determining this ratio; we present the polymer evolution under various reaction rates in Section Supplementary Information 2.2. This finding hypothesizes a central biophysical concept through our polymer model - reactions and diffusion compete to give rise to chromatin clutch domains, as observed primarily though super-resolution imaging- a result that has not been previously reported in an active, reaction-driven molecular dynamics setting.

**Chromatin domains exhibit nonlinear scaling with changing epigenetic reaction rates.** Recent advancements have highlighted the dynamic regulation of chromatin organization in response to chemo-mechanical cues from the cellular microenvironment, which influences the nuclear concentration of key epigenetic modifiers, mainly the histone writers and erasers[14–16]. The resulting epigenetic reactions, which capture shifts in the nuclear concentration of these remodelers, dictate the overall chromatin landscape by altering the heterochromatin-to-euchromatin ratio within the nucleus. The global effect of these reactions on chromatin organization is captured in Eq. (1), which we validate here. For this analysis, we fix the diffusion constant, focusing on understanding how the balance between methylation and acetylation reactions drives chromatin reorganization. We quantified the radius of gyration of repressed chromatin domains as a function of epigenetic reaction rates to gauge the extent of chromatin reorganization (Fig. 2d). As illustrated in Fig. 2d, the radius of gyration decreases as the acetylation rates increase (or the methylation rates decrease), confirming the analytical relationship in Eq. (1). Thus, we establish that as the number of acetylation (methylation) marks increases, there is a greater tendency for chromatin decompaction (compaction). This alteration in compactness can consequently affect chromatin accessibility in specific regions. The modified epigenetic marking and accessibility of chromatin alters potential interactions with the transcriptional machinery, thereby influencing gene expression. Hence, our model links changes in epigenetic modifier concentrations to chromatin organization and downstream effects via a single dimensionless factor—the ratio of the methylation rate to the acetylation rate ($\Gamma_{me}/\Gamma_{ac}$)—and demonstrates how the sizes of chromatin domains change with this factor.

**2c Data-informed chromatin polymer predicts spatial and epigenomic alterations in response to changing epigenetic reactions**

We initialize our simulations with experimentally derived epigenetic data, a critical step that significantly influences the subsequent chromatin reorganization dynamics. This represents a classic initial value problem, where the initial epigenetic state dictates the cell's epigenetic trajectory, shaping its future chromatin conformation and gene expression profile. Our model's robust generalization, rooted in its biophysical foundations as demonstrated in the preceding section where domains emerge in the prototype polymer, enables it to leverage sequencing sources such as ChIP-seq or Hi-C to guide the initial epigenetic distribution. As described in section Chromatin polymer model follows experimentally observed length scales and 119 incorporates diffusion and epigenetic reaction dynamics, we utilize Hi-C data for A375 cells (malignant melanoma with epithelial morphology) and ChIP-seq data for human mesenchymal stem cells (hMSCs derived

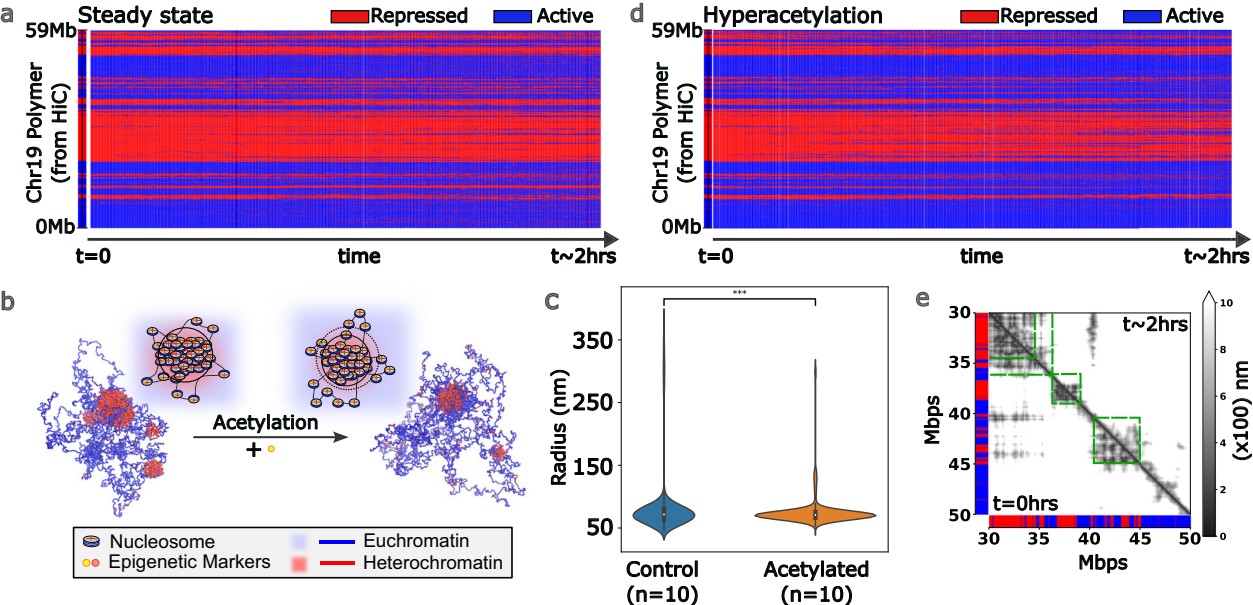

**Fig. 3 | Effects of changing the concentrations of histone epigenetic remodelers on chromatin. a** Temporal kymograph of chromosome 19 of A375 cells when the ratio of methylation to acetylation is the same as the initial heterochromatin to euchromatin ratio. The epigenetic domains temporally maintain their genomic extent on the polymer to a large extent (~90% for a single simulation). **b** With increasing acetylation, the repressed domain sizes decrease, as observed through the simulation snapshots. This finding was further validated through the observed radius of gyration of the chromatin domains in **c**. For 3 (**c**), Violin plots show data distribution. The center line marks the median, the box bounds the interquartile range (25th–75th percentile), and the whiskers span the minima and maxima.

Asterisks indicate statistical significance based on a two-sided $t$ test: $p < 0.05$ (*), $p < 0.01$ (**), $p < 0.001$ (***), and $p < 0.0001$ (****). **d** Kymograph evolution exhibiting euchromatin domains spread out to neighboring genomic regions as the number of acetylated beads increased. About 75% of the regions maintain their epigenetic marks. **e** The pairwise distance map shows the spatial reconfiguration of the polymer upon acetylation. The lower triangle corresponds to the initial pairwise distance between the polymer beads, and the upper triangle corresponds to the final pairwise distance. Upon acetylation, major decompaction is observed at the boundaries of heterochromatin domains, as shown by the increasing pairwise distance in the green boxes.

from human bone marrow) in the following sections. Our objective is to bridge the observed chromatin reorganization across mesoscale imaging and genomic sequencing, identifying potential genomic loci that are altered in response to changes in epigenetic reactions or the microenvironment.

In this section, by simulating the A375-informed chromatin polymer, we elucidate two key phenomena via our model: (i) the stability of chromatin epigenetic distribution in control conditions and (ii) the identification of genomic loci where epigenetic changes predominantly occur in response to changes in these rates.

**The data-informed polymer model preserves the chromatin domains with constant reaction rates.** To ensure that changes in chromatin organization are preserved under steady-state conditions, we examine the stability of chromatin domains when the reaction rates remain constant. Here, stability encompasses the epigenetic labeling of genomic loci and the spatial distance between chromatin segments, collectively influencing the cell's gene expression profile. The simulations are performed with $\varepsilon_{EE} = 0.3$ and $\varepsilon_{EE} = 0.3$. The reaction ratio, $\Gamma_{me}/\Gamma_{ac}$, is chosen to be the same as the initial heterochromatin-to-euchromatin ratio. The temporal kymograph in Fig. 3a (chromosome 19 of A375) shows the epigenetic marks at each chromatin segment (bead) as the simulation progresses. Remarkably, the initial distribution of epigenetic marks largely persists throughout the simulation, retaining its identity over a span of 2 h of real-time (extended simulations up to 10 h are illustrated in Supplementary Information 2.1), which demonstrates the ability of our model to maintain the epigenetic identity of individual chromatin segments. In addition, we

present the segment-to-segment distance matrix of the polymer at the start and end of the same simulation in Supplementary Information 2.1. The alignment between the initial and final pairwise distances of the chromatin segments further confirms the stability of the chromatin configuration as influenced by its dynamics. Thus, our model not only preserves the epigenetic identity but also maintains the accessibility of chromatin segments. Analogous analyses have been conducted for chromosomes 18, 20, and 21 in Supplementary Information 2.1.

**Modifying the reaction rates alters epigenetic marks at the boundaries of domains.** We next examine the changes in epigenetic marks of the polymer segments in response to variations in the reaction rates. We simulate a scenario where the acetylation reaction rate increases, resulting in a lower methylation-to-acetylation ratio (0.2) than the initial heterochromatin-to-euchromatin ratio (~ 0.45) for 2 h in real-time. This alteration in the reaction ratio led to a decrease in the size of the repressed chromatin domains, as evident from the reduced radius of gyration of these domains in Fig. 3b, which was further quantitatively confirmed in Fig. 3c. However, specific loci along the 1D polymer that exhibit alterations in epigenetic marks and accessibility as chromatin undergoes a transition to a more acetylated state can be identified in the case of data-informed polymers. The temporal kymograph of the polymer evolution (Fig. 3d) shows that the switch from the repressed to the active state is prominently localized to the domain boundaries of larger chromatin domains (~ 100 s of kbps) or smaller chromatin domains (few kbps) on the polymer. Our prediction shows that this heterochromatin-to-euchromatin conversion at the boundaries leads to chromatin decompaction, which is evident in the

pairwise distance map (Fig. 3e). Since boundary regions undergo epigenetic and spatial changes, these changes can potentially affect the transcriptional activity of genes at these genomic loci. We extended this analysis to other chromosomes in the A375 cell line, as presented in Supplementary Information 2.2. In Supplementary Information 2.2, we also show that an increased methylation rate changes genomic loci from euchromatin to heterochromatin, leading to chromatin compaction, which is also predominantly observed at domain boundaries.

This phenomenon of epigenetic marking alterations predominantly at domain boundaries arises from the fact that heterochromatin–heterochromatin interactions are significantly stronger than heterochromatin–euchromatin interactions, making it energetically favorable for epigenetic changes to occur at boundaries rather than within heterochromatin-rich domains, where they would require breaking favorable interactions. The mechanistic understanding of this phenomenon is as follows: Even though chromatin beads can undergo random changes in their epigenetic identity through epigenetic reactions. Once their identity changes, they tend to diffuse out of unfavorable environments and stabilize at domain boundaries, where the energy landscape is more favorable. We elucidate this mechanistic understanding of the boundary enrichment in Fig. 24 of the Supplementary Information. In conclusion, our dynamic model reveals a fundamental principle: chromatin regions at domain boundaries are most susceptible to changes in the concentration of epigenetic remodelers.

## 2 d Polymer model integrates complementary observations from super-resolution imaging and Hi-C sequencing, revealing epigenetic change specificity at domain boundaries

Next, we utilized super-resolution imaging (STORM) alongside bulk or population-level Hi-C sequencing to probe chromatin organization. Integrating these methods presents a challenge, as they offer complementary yet partial insights: STORM reveals the nanoscale organization of chromatin domains but lacks specific genomic loci data, whereas Hi-C supplies contact details for individual genomic loci but with limited spatial information. Furthermore, bulk techniques such as Hi-C overlook cellular heterogeneity, a factor captured by imaging. Our data-informed polymer model can predict both 3D morphological changes and associated genomic loci, thereby bridging this gap, and offering a comprehensive understanding of chromatin organization. We conducted super-resolution STORM imaging and bulk Hi-C sequencing on melanoma cells subjected to identical epigenetic modifications and analyzed the results with our polymer model to link these complimentary methods. This approach enabled us to elucidate the concomitant changes occurring at both the mesoscale (~100 s of kb scale) and the genomic scale (kb scale), providing a multiscale understanding of chromatin reorganization in response to changes in epigenetic reaction rates.

To investigate the effects of changes in epigenetic reaction rates on chromatin organization, we treated A375 melanoma cells with 0.5 μM trichostatin A (TSA), a well-characterized HDAC inhibitor, for 2 hours. This treatment is expected to increase acetylation levels, leading to a more decompacted, euchromatin-rich chromatin organization. We chose a 2 h timepoint to allow sufficient time for histone modifications to occur without secondary effects such as transcriptional changes influencing chromatin organization[68]. Immunofluorescence imaging of H3K9ac revealed a global increase in acetylation levels (Fig. 4a), while DAPI staining revealed significant changes in chromatin reorganization (Fig. 4a). To examine these changes at higher resolution, we performed STORM imaging, as described in the following subsection.

**STORM imaging shows qualitative agreement with model predicted chromatin domain scaling.** Super-resolution STORM imaging confirmed the presence of characteristic chromatin domains in A375

melanoma cells (Fig. 4b). Notably, following TSA treatment, the domain sizes significantly decreased across the two replicates (~5 cells each), as depicted by the representative STORM images in Fig. 4b. We quantified the domain sizes (detailed in Supplementary Information 3.1) across replicates and observed a statistically significant decrease in domain size (Fig. 4c). To validate these changes with our model predictions, we conducted simulations over a 2-hour real-time span, replicating the TSA treatment conditions by increasing the acetylation rate. These simulations were performed on chromosomes 18–21 of A375 cells. The simulated control and acetylated chromatin domains, as predicted by the simulation, demonstrated a marked decrease in the observed radius of gyration of the chromatin domains. The mean of the experimentally observed domain radii decreases by 8%, aligning closely with the simulation prediction of an 11% reduction. Furthermore, the size distributions of the experimental and simulated domains before and after the operation are similar, as illustrated in Fig. 4c.

**Averaging over the cell population leads to fewer epigenetic compartmental changes.** To explore the relationship between single-cell STORM and bulk Hi-C sequencing, we integrated population and single-cell effects into our simulations. At the single-cell level, kymographs illustrating the simulated effects of chromosome 19 acetylation in A375 cells (Fig. 3d) demonstrated domain shrinkage, suggesting a preference for compartmental shifts toward a more euchromatic configuration at domain boundaries. These findings indicate that epigenetic boundary regions are primarily responsive to changes in epigenetic remodeler activity. To quantify population-level changes, we performed multiple simulations with diverse initial spatial configurations of the chromatin polymer while maintaining Hi-C-informed epigenetic distributions. We introduced a flipping score to measure alterations in epigenetic marks, establishing a threshold to identify regions with significant changes (further elaborated in Supplementary Information 4.1). The flipping score assesses the proportion of cells exhibiting a compartmental shift, with the threshold set to detect significant changes present in the majority (>50%) of observed cells. This approach facilitates the integration of cellular heterogeneity into our analysis, enabling us to consider its influence on population-level observations. As a result, we gain a more nuanced understanding of chromatin organization and its variability across cells.

As we averaged over simulations, fewer regions passed the threshold for compartment change (Fig. 4d, i), demonstrating how cellular heterogeneity is reflected in bulk experiments. As presented in Fig. 4d and i, averaging over multiple simulations leads to the suppression of domain changes, which are specific to only a subpopulation (corresponding to averaging over a few simulations), and reveals only those changes that are present more frequently in the whole population (which corresponds to averaging over multiple simulations). Notably, chromatin segments with high flipping scores at domain boundaries survived such an averaging operation (Fig. 4e), as they corresponded to changes in most of the cells. We elucidate this general observation through the schematic in Fig. 4f, which illustrates that heterogeneous cell-to-cell differences are more prominent in single-cell observations than in bulk sequencing observations, where population averaging obscures these variations. The schematic highlights how population-wide changes, such as those at the boundary domains that are present for most of the cells, are captured in the Hi-C observations (or other bulk sequencing observations), but other subpopulation-level changes are not. Our analysis highlights the importance of considering cellular heterogeneity in single-cell vs. bulk experiments, as bulk experiments such as Hi-C sequencing good at highlighting the significant population-wide changes. To validate our genomic-scale predictions, we turn to bulk Hi-C experiments, which provide a means to robustly quantify compartment changes.

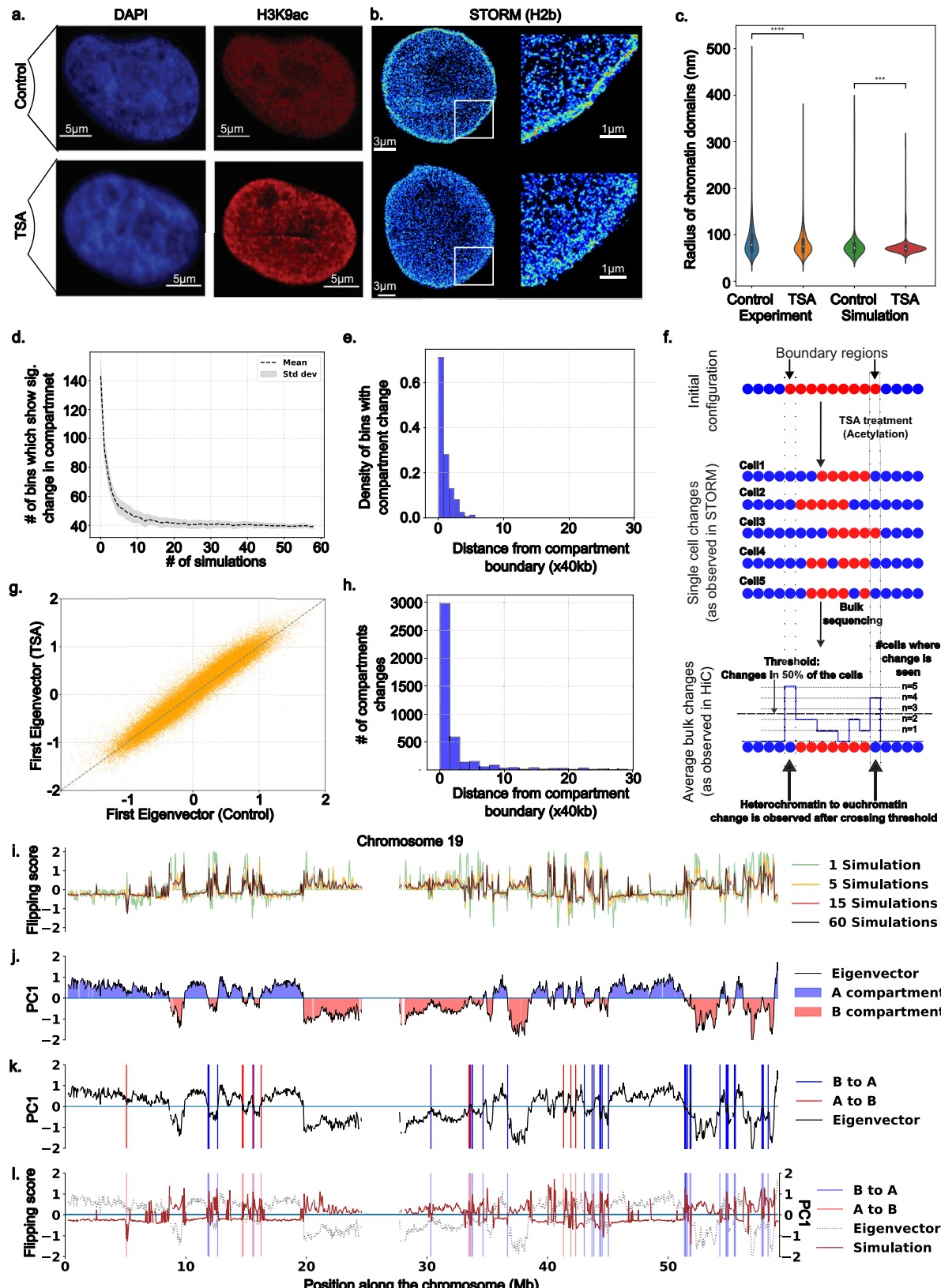

**Changes in bulk Hi-C contact maps reveal epigenetic compartment shifts at domain boundaries.** We generated Hi-C maps for two replicates of control and TSA-treated A375 cells, revealing only minor rearrangements (Supplementary Information 4.1) despite substantial alterations in domain size scaling under STORM imaging and disparate H3K9ac immunofluorescence levels. To systematically assess changes

in the Hi-C maps, we combined the two Hi-C matrices at 40 kb resolution and employed principal component analysis (PCA) to delineate A (active, euchromatin-like) and B (inactive, heterochromatin-like) compartments for both control and TSA-treated cells (Fig. 4j) (details on Hi-C analysis in Supplementary Information 4.1). Comparative PC1 analysis revealed that most regions retained their compartment

**Fig. 4 | Chromatin polymer model bridges STORM and Hi-C observations. a** TSA treatment of A375 melanoma cells shows global chromatin reorganization (DAPI, blue) and increased nuclear acetylation (H3K9ac, red). **b** STORM images (H2B-labeled) reveal that TSA-treated cells exhibit decompacted chromatin with smaller interior domains compared to controls. **c** Simulations replicate these domain-size trends by increasing the acetylation rate, mirroring the hyperacetylated state. Experimental data spans two replicates (five cells each), and simulations include ten runs for chromosomes 18–21. In the violin plots, the center line marks the median, box bounds indicate the interquartile range, and whiskers span the minima and maxima. Asterisks denote significance from a two-sided $t$ test: $p < 0.05$ (*), $p < 0.01$ (**), $p < 0.001$ (***), $p < 0.0001$ (****). **d** To approximate population-level data, multiple single-cell simulation runs are averaged. For chr19, the fraction of 10 kb beads flipping compartments decreases threefold after ~30 simulations and plateaus.

**e** Most flips occur near domain boundaries, reflecting their susceptibility to epigenetic remodeling. **f** A schematic shows that individual cells may shift a heterochromatin domain from nine to five beads, but averaging yields a size of seven, capturing majority trends while masking heterogeneity. **g** Principal component (PC1) analysis of TSA vs. control Hi-C (40 kb bins) shows limited global compartment changes. **h** Approximately 70% of compartment changes occur within 40 kb of boundaries, indicating boundary-driven epigenetic shifts. **i** Simulation flipping scores ($\pm 2$) measure the probability of switching from B to A (positive) or A to B (negative); values beyond $\pm 1$ are significant. **j** The A/B compartment profile from the PCA of control Hi-C contacts serves as a baseline. **k** Observed B → A (blue) and A → B (red) changes in Hi-C data. **l** Overlaying simulation-based flipping scores with experimental changes confirms strong agreement, underscoring the model's accuracy in predicting boundary-focused chromatin reorganization.

identity, with no significant shifts in epigenetic patterns (Fig. 4g). Among approximately 70 k mappable bins at 40 kb resolution, only about 5k (~7%) bins exhibited changes in genomic compartments. Notably, among these 5k bins, nearly 3.1k bins switched from B to A compartments, whereas ~1.3 k shifted from A to B compartments. We defined bins that transitioned from negative to positive PCA values as those that shifted from B-like to A-like behavior, and vice versa. This trend persisted when a threshold based on compartment strength was applied (TSA PC1 < − 0.01 for bins going from A to B and TSA PC1 > 0.01 when going from B to A), yielding 2838 bins switching from B to A and 1148 bins switching from A to B. Although these changes are relatively small in the context of the entire genome, they affirm our key predictions: regions tend to transition from repressed to active states upon TSA treatment. The observation that more noticeable changes in STORM imaging correspond to smaller changes in Hi-C contact patterns confirms our previous discussion on bulk and single-cell observations. Our model clearly shows that significant changes in single-cell STORM imaging correspond to subdued changes in Hi-C data due to averaging. Next, we analyzed the precise genomic locations of these compartment switches.

We examined the initial epigenetic characteristics of the compartments that underwent state changes, revealing that nearly half of the bins that shifted compartmental signatures had initial PC1 absolute values of 0.1 or less. These findings suggest that these regions are initially less strongly associated with their specific epigenomic domains, increasing their susceptibility to alteration upon epigenetic perturbation. To investigate the genomic locations of these altered bins, we plotted their positions on the genome. Figure 4k shows the genomic positions of bins transitioning their epigenetic compartment from A to B (marked by blue vertical lines) and from B to A (marked by red vertical lines) for chromosome 19 of A375 cells. This visual representation reveals that the observed compartmental shifts predominantly occur very close to the domain boundaries. To quantify this result, we analyzed this phenomenon genome-wide, which revealed that most compartment switches (over 75%) are located within 50 kb of boundary regions (Fig. 4h). This finding confirms that compartmental boundaries are notably responsive to changes in epigenetic reactions. When we overlay these experimental changes with the predictions generated by our polymer model after averaging over multiple simulations (Fig. 4l), a clear alignment emerges, with high flipping scores aligning with positions where a compartment change is observed in Hi-C.

In summary, our model highlights two key results: (1) While we observe noticeable changes in STORM imaging, the Hi-C data reflect only those changes that are present for the majority of the cells while not capturing changes that are shown by smaller subpopulations. (2) The compartment boundaries, as confirmed by Hi-C observations, emerge as regions of heightened sensitivity to epigenetic reaction alterations, where transitions in chromatin compartmentalization are favored owing to the enacting kinetics. We next analyze the downstream effects of these epigenetic changes via RNA-seq analysis.

## 2e RNA-Seq of A375 cells revealed key differentially expressed genes near domain boundaries that are pivotal for epithelial-to-mesenchymal transition and metastasis

To analyze the changes in gene expression that accompany chromatin reorganization, we performed RNA-seq on 2 h TSA-treated A375 cells. Among the significantly differentially expressed (DE) genes ($|log2(TSA/Control)|>1$ and $p$-value < 0.05), 423 genes were upregulated, and 124 genes were downregulated (Fig. 5a). This finding is consistent with our model prediction that HDAC inhibition leads to increased euchromatin content and thus can cause preferential global upregulation of genes. Analysis of the RNA-seq data with clusterProfiler[69] via gene ontology (GO) overrepresentation analysis (ORA) for biological pathways revealed that among the differentially regulated pathways, the most significant were those related to cell migration (Fig. 5b, top panel). Since A375 cells are malignant melanomas with an epithelial morphology, this observation suggests an altered metastatic potential for cells treated with TSA[70]. We also found that the *WNT* signaling and *MAPK/ERK* pathways were significantly differentially regulated (Fig. 5b, bottom panel). A large body of published work shows that *WNT* (canonical and noncanonical) signaling[71–73] and *MAPK/ERK* signaling[74–76] are instrumental in driving the epithelial-to-mesenchymal transition (EMT). This could translate to an altered metastatic tendency in A375 cells through EMT[77–83] upon TSA treatment and, therefore, be differentially regulated with changes in the tumor microenvironment. Our findings further confirm the widely recognized phenomenon that HDAC inhibition leads to altered metastatic potential by differentially regulating key pathways[84–89].

We integrated RNA-seq data, model predictions, and Hi-C data by analyzing genes in genomic bins with compartment changes. We segregated all genes into two groups: those in bins transitioning from A to B compartments and those in bins transitioning from B to A (Fig. 5c). Our analysis revealed a significant preference for upregulation among genes transitioning from B to A but no discernible trends for genes transitioning from A to B. This finding supports our model's predictions, suggesting that hyperacetylation-induced decompaction at domain boundaries enhances gene upregulation. Notably, we found that differentially regulated genes at domain boundaries are highly enriched for positive regulation of epithelial cell migration (GO:0010634, FDR = 3.3e-05), indicating that epigenetic remodeling at domain boundaries may play a key role in regulating this critical melanoma metastasis process.

Since gene regulation can be driven by epigenetic changes several kilobases away from the coding region, we categorized all the differentially expressed genes within 300 kb as boundary genes (~30% of DE genes) and the remaining genes as genes located away from the domain boundary. We performed GO-ORA analysis for the genes within 300 kb, which revealed that pathways associated with epithelial-to-mesenchymal transition were enriched in this gene set (Supplementary Information 4.2). To ascertain that this is not a general trend among any subset of DE genes and is specific to genes close to the domain boundaries, we compared the enrichment of the EMT gene

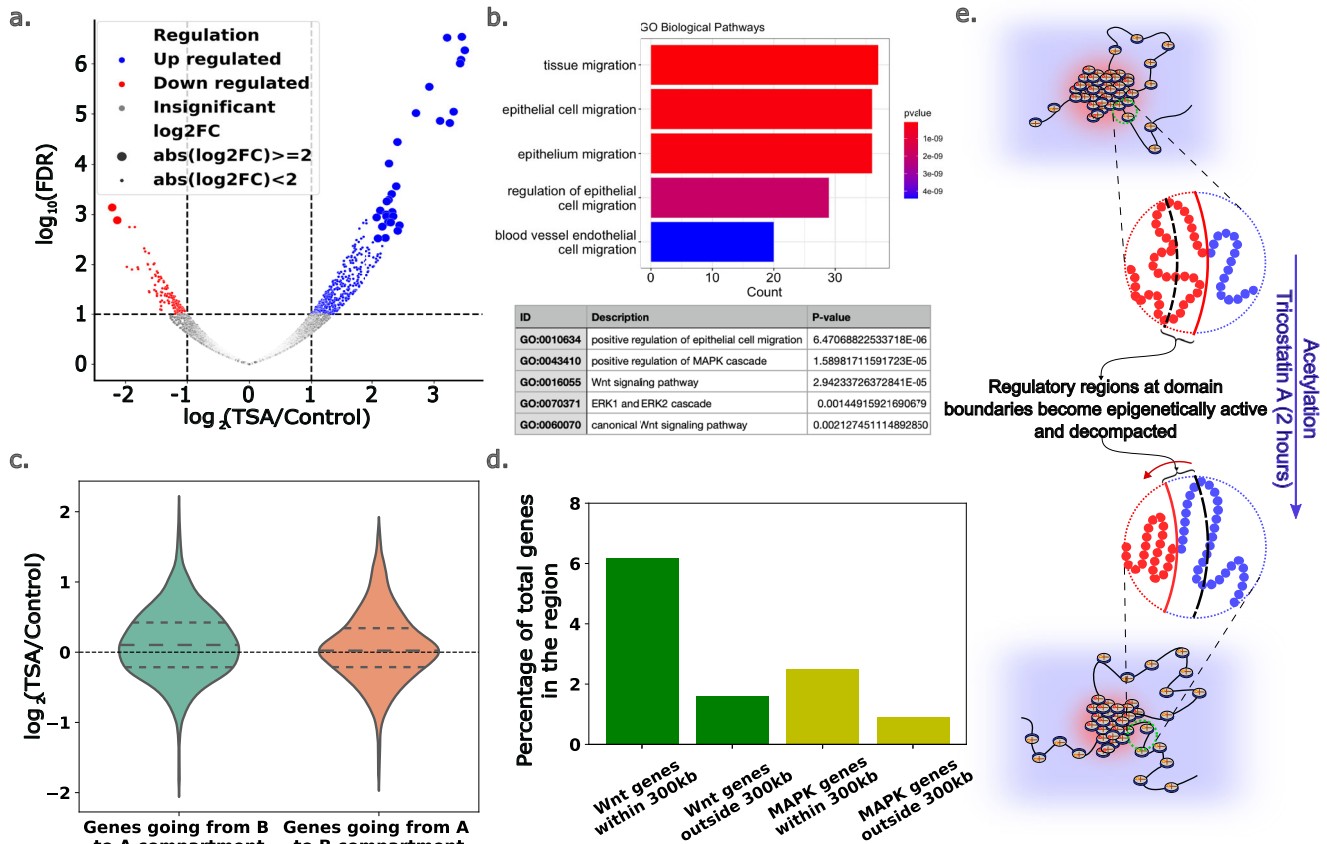

**Fig. 5 | Epigenetic changes at domain boundaries can play a central role in determining cell fate. a** Global RNA-seq trends showing that among the significantly differentially expressed genes, ~420 genes are upregulated and ~120 are downregulated, indicating that, upon HDAC inhibition (TSA treatment), genes are preferentially upregulated globally. **b** Overrepresentation analysis of biological pathways of the upregulated genes via clusterProfiler shows that cell migration is significantly altered after treatment. Over-Representation Analysis (ORA) used Fisher's exact test (one-sided), with *p*-values adjusted via the Benjamini-Hochberg (BH) method to control FDR. Moreover, the canonical WNT signaling and MAPK/ ERK pathways were significantly upregulated. These pathways are particularly important for epithelial-to-mesenchymal transition and are, therefore, central to metastasis. **c** Among all genes belonging to genomic bins where an epigenetic shift is observed in the Hi-C data, genes belonging to the bins with B going to the A compartment show a significant upregulation trend, whereas the opposite compartment flip has no discernable trend. **d** Focusing on genes that are differentially regulated from the WNT and MAPK pathways, we show that they are significantly enriched closer to the domain boundaries than away from it, suggesting that epigenetic changes at the domain boundaries may drive these changes. **e** Schematic showing how decompaction and epigenetic transformation at domain boundaries expose regulatory regions. This allows for differential expression of the pathways associated with genes/genes with regulatory elements (enhancer, promoter, etc.) in this region.

sets close to the domain boundary against those away from the boundary. Specifically, we focused on EMT-relevant GO pathway subsets (*WNT* and *MAPK/ERK*) that were enriched in the global subset. For these pathways, we plotted their gene ratios within the 300 kb region and outside the 300 kb region (Fig. 5d). We found that both pathways are significantly overrepresented closer to the domain boundary than away from it, suggesting that perturbations to the epigenetic domain boundaries can drive changes in their expression (gene list provided in Supplementary Information 4.2). This analysis revealed that perturbations in nuclear epigenetic remodelers, such as those of HDACs, not only epigenetically drive changes enriched at domain boundaries but also potentially lead to differential regulation of critical genes in these regions.

In summary, this section elucidates the pivotal role of genes near domain boundaries in orchestrating cellular behavior by transitioning between active and inactive states (depicted in Fig. 5e). Our analysis reveals a robust, overarching mechanism for modulating the cellular phenotype through chromatin reorganization. Importantly, this mechanism facilitates cell line- or history-dependent alterations in cellular behavior resulting from the significant influence of preexisting

epigenetic landscapes on phenotypic state changes. Thus, in addition to specific transcriptional pathways governing gene networks, this global mechanism can engender cellular heterogeneity based on a cell's developmental history. While our demonstration focuses primarily on these changes in response to TSA treatment, we broaden the scope to underscore the wider applicability of our model and demonstrate the versatility of the framework we have established.

## 2 f Polymer model predicts chromatin domain size distribution after microenvironmental stiffness changes

In the previous sections, we established the occurrence and scaling of the chromatin domains as predicted by the polymer model in response to changing nuclear epigenetic remodeler concentrations and established our findings through HDAC inhibition via STORM and Hi-C sequencing. However, nuclear chromatin remodeler concentrations can also change in response to changes in the mechanical properties of the microenvironment. Specifically, it has been shown extensively[23,25,26] that changes in extracellular stiffness alter the concentrations of nuclear chromatin remodelers, specifically HDAC3 and EZH2 (which are histone methyltransferase/HMT). In this section, we show that our

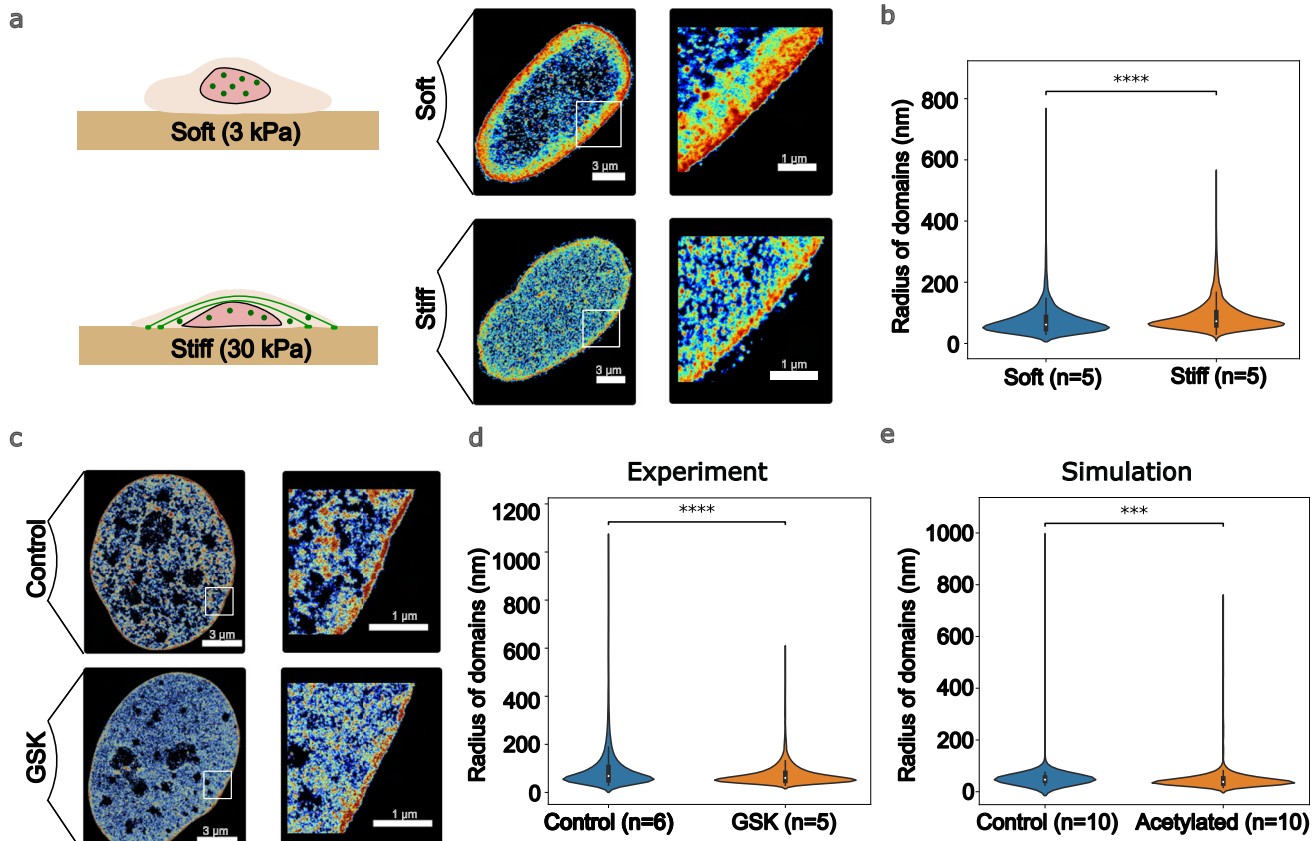

**Fig. 6 | Polymer model captures chromatin reorganization in response to changes in microenvironmental stiffness. a** hMSCs are cultured on 3 kPa (soft) and 30 kPa (stiff) substrates. H2b is labeled and imaged. The chromatin domains decrease in size on a stiffer substrate because of changes in the concentrations of nuclear epigenetic remodelers (represented by green dots)[23]. **b** An increase in substrate stiffness drives the decompaction of chromatin, resulting in a decreased observed radius of the chromatin domains. **c** GSK treatment of hMSCs inhibits EZH2 (an HMT), leading to lower methylation levels and, thus, chromatin decompaction, as shown in the STORM images. **d** GSK treatment decreases the observed domain sizes, captured through the observed radii of the clutches. **e** Simulation of the hMSC data-driven polymers for chromosomes 19–22 replicates the observed decrease in domain size. For **b**, **d**, and **e**, violin plots show data distribution. The center line marks the median, the box bounds the interquartile range (25th–75th percentile), and the whiskers span the minima and maxima. Asterisks denote significance from a two-sided *t* test: $p < 0.05$ (*), $p < 0.01$ (**), $p < 0.001$ (***), $p < 0.0001$ (****).

model can be extended to super-resolution imaging experiments published by Heo et al.[23], where chromatin architecture remodeling of a human mesenchymal stem cell (hMSC) is observed in response to changes in microenvironment stiffness. For this analysis, we used hMSC chromosomes 19-22 data-informed polymers constructed from histone ChIP-seq data. The current model predicts chromatin reorganization based on observed changes in epigenetic reaction dynamics in response to variations in substrate stiffness.

**Chromatin domains decondense on a stiffer substrate in hMSCs.** Heo et al.[23] cultured human mesenchymal stem cells (hMSCs) on methacrylated hyaluronic acid (MeHA) hydrogels with distinct stiffnesses (30 kPa and 3 kPa) for 48 h (Fig. 6a). STORM imaging with H2b histone protein labeling revealed a global view of the spatial organization of the genome, revealing distinct chromatin domains with significantly different size distributions between the two samples (Fig. 6b). The softer substrate featured larger domains with more compacted chromatin, whereas the stiffer substrate displayed smaller domains with more decompacted chromatin (more details are provided in Supplementary Information 3.1). A relative reduction of 8% was observed in the domain sizes from soft to stiff substrates. Histone tail mark analysis revealed a substrate stiffness-dependent increase in H3K4me3 and a decrease in H3K27me3, indicating epigenetic regulation of chromatin reorganization. Treatment with the ROCK inhibitor

Y-27632 abolished the substrate stiffening effect, confirming that enhanced cellular contractility drives this global increase in nuclear acetylation and chromatin decompaction (data in Supplementary Information 3.2). Consequently, hMSC chromatin reorganizes into a more decompacted state on stiffer substrates owing to altered cellular contractility.

**Chemically induced HMT inhibition leads to chromatin domain decompaction.** To ascertain whether H3K27me3 changes can also induce chromatin reorganization, EZH2, a histone methyltransferase crucial for H3K27me3 catalysis, was inhibited by Heo et al.[23] Using GSK343 (GSK), a specific EZH2 inhibitor, it is confirmed that chromatin structure alterations result from epigenetic profile variations mediated by epigenetic remodelers. Previous studies[23] have shown that on stiffer substrates, EZH2 relocates out of the nucleus, increasing the proportion of euchromatin due to decreased H3K27me3 levels. It is expected that GSK treatment would yield a similar trend as switching to a stiffer substrate. Human mesenchymal stem cells (hMSCs) were treated with GSK throughout their two-day culture period, and subsequent STORM imaging revealed significant chromatin decondensation, characterized by a reduction in condensed chromatin domain size (marked by an 18% reduction in domain size upon treatment) (Fig. 6c, d) (more details are provided in Supplementary Information 3.1). This finding confirms that the H3K27me3 changes observed with changing substrate stiffness are

sufficient to drive chromatin reorganization. Thus, this presents an ideal scenario for the application of our model, where microenvironmental cues govern chromatin remodeling through nuclear epigenetic tuning.

**The polymer model captures changes in chromatin organization in response to mechanical cues, specifically stiffness.** We validated our model predictions by comparing them to experimentally observed chromatin reorganization trends. Since changes in the concentrations of nuclear epigenetic remodelers, specifically EZH2, drive chromatin reorganization in response to both stiffness changes and HMT inhibition (GSK treatment), we simulated this phenomenon by adjusting the epigenetic reaction rates, which capture the effect of EZH2 in our model. Control simulations were run with a constant net epigenetic makeup, whereas simulations with decreased methylation levels (representing increased stiffness and GSK treatment) were run with adjusted acetylation-to-methylation reaction rates (0.2). The resulting domain size distributions (Fig. 6e) strongly resemble the experimental observations, with a simulated reduction in the mean domain radii of 15%, which aligns with the experimental trends.

Finally, our polymer model can be integrated with our previously published models, linking actin polymerization to nuclear epigenetic remodeler concentrations[25,26]. This hybrid framework enables the prediction of chromatin organization changes in response to alterations in the cellular cytoskeletal network across diverse microenvironments. By combining epigenetic marks and chromatin organization changes, our model captures the genomic effect of mechanical alterations in tissue properties that occur in diseased states, such as tendinosis. Notably, our model reveals that domain size distribution changes in tendinosis following the chromatin reorganization principles we revealed, replicating the observed chromatin domain scaling trends in cases where nuclear histone epigenetic remodelers play a key role (data in Supplementary Information 3.2)[23]. This phenomenon is particularly significant in response to changes in microenvironmental stiffness, highlighting the model's potential to elucidate the complex interplay between mechanical and epigenetic regulation in disease.

## 2 g Polymer model uncovers a robust mechanism of microenvironment memory formation

Epigenetic memory becomes particularly relevant when the cell passes through multiple microenvironments and has the potential to experience lasting influences from these environments. This is important for processes such as metastasis, wound healing, or morphogenesis, where the cells go through rapidly changing environments, raising the question of how and where the memory of the microenvironment is being stored. Recent work[54] has shown how 3D organization, along with a limitation in epigenetic modifying enzymes, can lead to the efficient passage of epigenetic memory from one generation to the next. Our model provides an opportunity to examine how exposure to an environment affects the cell phenotype during the cell cycle. To address this aspect through our model, we take the example of chromosome 19 in the hMSC cell line, which is mesenchymal in nature, with a plastic phenotype compared to fully differentiated cells. To understand the long-term behavior in a varying microenvironment, we start by taking the active-repressed compartmentalization as dictated by the ChIP-seq data (section Chromatin polymer model follows experimentally observed length scales and 119 incorporates diffusion and epigenetic reaction dynamics). We then increased the acetylation for time t′, followed by an increase in methylation for time t′, amounting to a total simulation of 2t′. In an experimental setting, this would correspond to the stiffening and subsequent softening of a substrate (with changes in nuclear HDAC, as shown in section Data-informed chromatin polymer predicts spatial and epigenomic alterations in 280 response to changing epigenetic

reactions) or to TSA treatment with subsequent washout. In the plotted simulation, the time t′ is chosen to be ~ 5 hours of real-time, and the acetylation-to-methylation ratio is chosen to be 0.8 for t′ and reduced thereafter, as depicted in the kymograph in Fig. 7a. We observe that the domain sizes decrease with acetylation, followed by an increase with increased methylation (Fig. 7b).

To analyze the memory-like behavior in our system, we look at 3 different kinds of polymer beads, according to their epigenetic trajectory (Fig. 7c): (i) we refer to beads that never change their marks as "conserved beads," (ii) the beads that change their epigenetic character upon exposure to acetylation (stimuli) and never switch back are referred to as "memory from the shock" and (iii) "recovered" beads are those that change their epigenetic mark but return to their original epigenetic mark after the stimulus is removed. These three types of regions on the chromatin thread highlight the major behaviors one would expect to see in the cell when undergoing extracellular stimulation: (i) some phenotypes that are "conserved," (ii) another set of phenotypes that switch permanently based on external cues and, finally, (iii) a set of phenotypes that change elastically with the microenvironment. We observe that the extent of these regions is sensitive to the magnitude of extracellular stimulus, with more irreversible changes (higher proportion of memory beads) as the external cues increase (Fig. 7d). Detailed kymographs are plotted in Supplementary Information 2.3.

In addition to the central role of the sensitivity of the domain boundaries in response to external cues, we find that regions flanking the domain boundaries play a central role in determining the eventual epigenetic and, therefore, transcriptional state of the chromatin. In this context, we subcategorize the domains into large domains and smaller domains (as shown through the green and cyan panels of Fig. 7a) to help us understand their evolution better. For the larger domain, we show the corresponding evolution in Fig. 7e. We have three different regions, Regions I, II, and III. Both Regions I and III, which are spatially far from the domain boundaries, maintain their epigenetic and, therefore, accessibility characteristics. Region II, which is at the boundary, undergoes an epigenetic and accessibility cycle throughout the simulation. Through these changes, this region does not recover fully to its original state and contributes to both the formation of permanent epigenetic memory and epigenetic recovery. In contrast, all regions of smaller domains (cyan panels in Fig. 7a and f), which are spatially in the vicinity of domain boundaries, are destroyed by shifts in the epigenetic makeup (Fig. 7f, Region I) and new domains are formed through a nucleation and growth process (Fig. 7f, Region II). Therefore, smaller domains can act as domain boundaries themselves and give rise to lasting memory characteristics.

Through this analysis, we hypothesize the central role of domain boundaries in the establishment, maintenance, and dissipation of extracellular memory. Since we observe interesting plastic dynamics even with a coarse-grained system and two epigenetic flavors; in a more realistic system, multiple epigenetic states and their differential propensity to switch can give rise to highly complex cellular phenotype evolution dynamics.

## Discussion

Over the past decade, revolutionary advances in experimental techniques, particularly super-resolution imaging (STORM[90], PALM[91], ChromSTEM[92], etc.) and next-generation sequencing (Hi-C[5], ChIP-seq[13], ATAC-seq[93], etc.), have transformed our understanding of the three-dimensional organization of the mammalian genome and its role in determining the transcriptional state of the cell. Concurrently, significant strides in the analytical and computational modeling of chromatin[94–97] have established a biophysical framework for understanding chromatin packing mechanisms and the pivotal role of epigenetic markers and looping in driving this organization. These

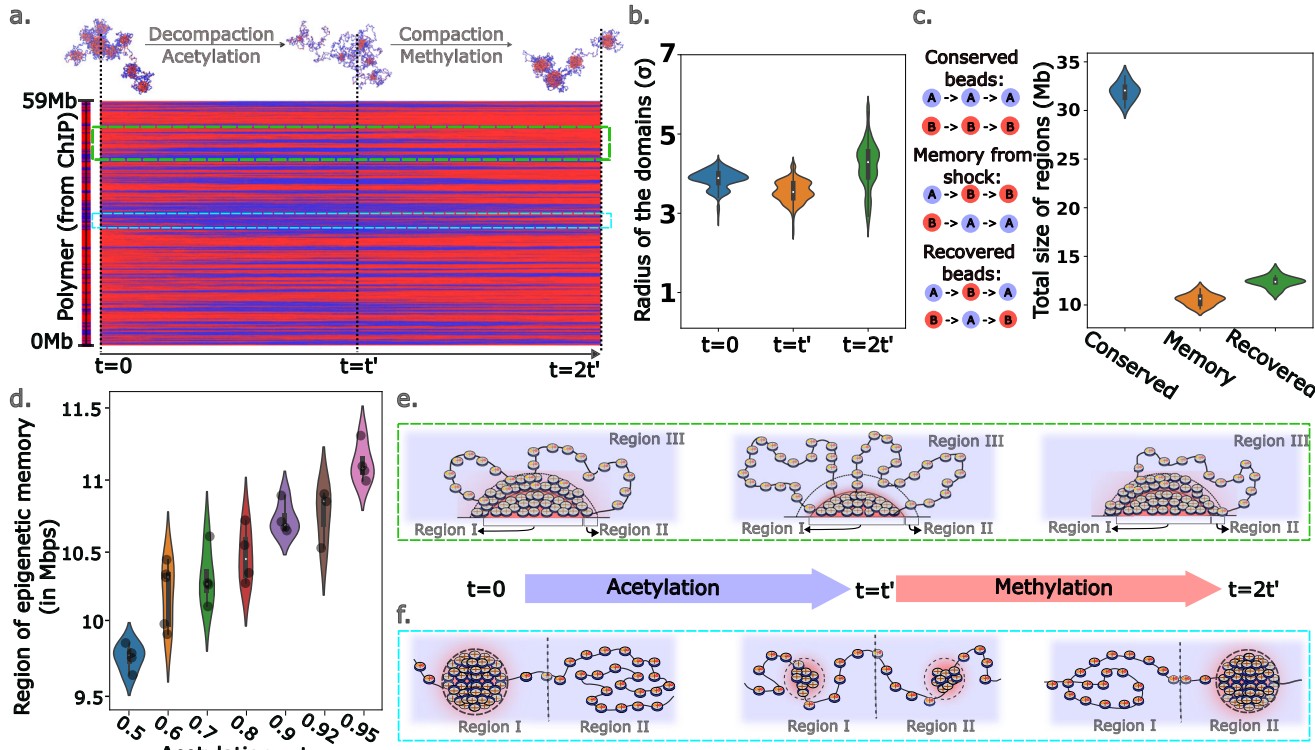

**Fig. 7 | Polymer model reveals a robust mechanism for chromatin organization-driven epigenetic memory.** **a** Initial acetylation of chromosome 19 from hMSCs (from $t=0$ to $t=t'$) followed by methylation ($t=t'$ to $t=2t'$). This process is accompanied by simultaneous chromatin decompaction and then compaction. **b** Change in the chromatin radius upon acetylation and subsequent methylation averaged over 10 simulations. The domain sizes first decrease in size because of decompaction resulting from acetylation and then grow because of compaction due to methylation. **c** Evolution of the chromatin is shown through three types of beads: conserved, memory, and recovered (averaged over 10 simulations). Conserved beads do not change their epigenetic marks throughout the simulation time. Beads belonging to the memory from the shock category switch their epigenetic mark during acetylation but do not recover during methylation. Recovered beads lose their original epigenetic mark during acetylation but regain it during methylation. Their total size on chr19 is plotted based on multiple final polymer configurations. **d** Dependence of the total memory bead segment is plotted as a function of the extent of acetylation (4 simulations for each case). Higher acetylation leads to

higher memory of the intermediate state in the same time frame. **e** Schematic showing the behavior of large heterochromatin domains through the simulation. The corresponding simulation region is shown in the green box in subfigure (**a**). The beads in regions I and III are largely conserved, and their epigenetic marks are maintained throughout the simulation. Beads in region II contribute to recovered beads and memory beads, switching their epigenetic marks during acetylation but regaining their original marks or switching and not coming back. **f** Schematic showing the behavior of smaller heterochromatin domains through the simulation. The corresponding simulation region is shown in the cyan box in subfigure (**a**). Region I loses its epigenetic marks during acetylation and decreases in size. Another neighboring region, Region II, forms a nucleating heterochromatin domain. As the methylation rates increase, region II grows to form a distinct domain. For **b** and **d**, violin plots show data distribution. The center line marks the median, the box bounds the interquartile range (25th–75th percentile), and the whiskers span the minima and maxima.

breakthroughs have elucidated the multiscale structure of chromatin, spanning from chromatin packing domains[21], visualized through super-resolution imaging, to compartments, topologically associated domains (TADs), and loops, which have been brought to light, primarily through chromatin conformation capture mapping techniques[98]. Data-intensive molecular dynamics approaches[99] and subsequent deep learning efforts[100] have successfully predicted chromatin capture maps via DNA sequences and epigenetic tracks as inputs, demonstrating a strong correlation of compartmentalization with epigenetic marks and their associated proteins. Furthermore, integrated experimental-modeling approaches have provided insight into the dynamic reorganization of chromatin during the transition from interphase to mitosis[101], revealing the crucial roles of looping motors[48] and transcription-driven supercoiling dynamics in chromatin compaction and reconfiguration[11,102,103].

Despite significant progress in understanding the relationships between chromatin organization and the epigenetic machinery, the effects of far-from-equilibrium epigenetic remodeling processes on chromatin structure and function remain comparatively less understood. Among the various contributing factors to out-of-equilibrium chromatin dynamics, cohesin- and condensin-dependent looping

mechanisms are very well understood[48]. The fact that cohesin depletion leads to a noisier expression pattern rather than significantly altered gene expression suggests the importance of other mechanisms at play[104]. In this light, the role of epigenetic-reaction-dependent chromatin domains, which persist even after Rad21 depletion[105], prominently observed through super-resolution imaging, is worth exploring. Therefore, in the current work, we aimed to explore the role of histone epigenetic remodelers (e.g., HDACs and HMTs) on these chromatin domains in dictating chromatin organization. Efforts to understand other epigenetic remodeling mechanisms, such as the spreading of epigenetic marks across cell replication cycles[44,53,54], have been carried out, but to the best of our knowledge, no molecular dynamics-based polymer model has explicitly considered diffusion and microenvironment-driven epigenetic reactions to explain chromatin organization through the formation of heterochromatin-rich packing domains. Crucially, current models are not able to explain the ability the methylation and acetylation reactions in the formation and size scaling of chromatin domains in different scenarios, including disease, development, aging, and the cellular response to drugs. In this study, we take a hybrid approach, integrating simulations with sequencing and imaging experiments to address this knowledge gap.

Our molecular dynamics-based model has the following key distinctive features:

1. We employ a data-driven methodology, utilizing next-generation sequencing to inform epigenetic labeling of the chromatin polymer. This enables the design of in silico experiments that can be directly compared with alterations in chromatin architecture resulting from changes due to extracellular cues, ranging from mechanical cues, such as substrate stiffness, to the impact of pharmacological and epigenetic drugs. This facilitates a robust validation framework to test the model.

2. Our model incorporates the impact of changes in histone methylation and acetylation on chromatin organization by modeling histone epigenetic remodelers through epigenetic reactions. This feature enables the simulation and analysis of chromatin architectural changes in response to various stimuli within a unified framework.

3. Through chromosome-scale simulations, our model bridges scales by capturing chromatin structure at the nanoscale while simultaneously tracking alterations across multiple kilobases. This allows us to integrate STORM imaging observations with local perturbations detected by next-generation sequencing, providing a unified understanding of chromatin organization.

In our far-from-equilibrium chromatin dynamics model, euchromatin and heterochromatin segments are assigned via binarized ChIP-seq or Hi-C data. To account for the interplay between epigenetics and the chromatin configuration, our model accounts for energetically driven chromatin–chromatin interactions leading to phase separation of epigenetic domains through Brownian motion and epigenetic diffusion. The out-of-equilibrium dynamics are introduced through epigenetic reactions that capture the action of histone epigenetic remodelers and control the active interconversion of heterochromatin and euchromatin states. Recent work[23,25,26] has shown that these epigenetic remodelers are highly sensitive to the microenvironment. Hence, through the current model, we predict the resulting epigenetic landscape and chromatin accessibility, given the initial epigenetics and the variations in the extracellular-driven nuclear concentrations of histone epigenetic modifiers.

Using this epigenetic reaction-driven approach, we demonstrate that the proposed dynamics capture the formation of characteristic nanoscale heterochromatin-rich domains. Our model reveals that heterochromatin-rich packing domain formation results from competition between passive diffusion and active epigenetic reactions, wherein the passive component drives the system toward Ostwald ripening, whereas the active reactions oppose this through energetically unfavorable interchange of epigenetic marks. This competition leads to the scaling of the domain radii with $\sqrt{D/\Gamma_{ac}}$, where D is the diffusion constant and $\Gamma_{ac}$ is the acetylation reaction rate. We have also shown that the radius of chromatin domains changes nonlinearly with changes in epigenetic reactions, with the radii of the domains scaling proportional to $\sqrt{\Gamma_{me}/\Gamma_{ac}(\Gamma_{me}+\Gamma_{ac})}$, capturing the changes in the chromatin organization with changes in the histone epigenetic remodeler concentrations. We confirmed this size-scaling behavior of the chromatin domains through super-resolution imaging of unperturbed and hyperacetylated A375 melanoma cells via TSA (HDACi) treatment. While our study demonstrates robust evidence for the scaling behavior of chromatin domains, testing the full ripening dynamics remains beyond current technological capabilities. Decisively validating our model would require inhibiting all epigenetic reactions simultaneously to observe chromatin behavior, but this is not feasible in a living cell.

Next, we utilized our polymer model to investigate kilobase-scale changes in response to changes in epigenetic reaction rates. Our model reveals that, at the single-cell level, while changes in epigenetic reactions lead to genome-wide epigenetic and conformational perturbations, they are more likely to occur at the kilobase-scale

boundaries of heterochromatin domains. When observed over a population of cells, analogous to bulk sequencing of heterogeneous cell populations, our simulations show that these epigenetic shifts progressively concentrate at domain boundaries. These findings suggest that population-level sequencing methodologies such as Hi-C and ChIP-seq are likely to overlook cell-to-cell heterogeneity, which single-cell techniques can capture. We find that imaging techniques, and in general single-cell techniques, can reveal drug exposure changes that are not robust across populations. On the contrary, bulk sequencing technologies excel at obtaining dominant population-wide changes. To validate our epigenetic switching predictions, we performed Hi-C sequencing on hyperacetylated melanoma cells and observed concentrated epigenetic shifts at domain boundaries without widespread compartmental changes. RNA-seq analysis revealed a correlation between epigenetic landscape alterations and transcriptional state changes in melanoma cells, with boundary region genes following local epigenetic and accessibility trends. Notably, key EMT-related genes near domain boundaries were upregulated, highlighting the crucial role of domain boundaries in determining cell fate. Our predictions were further validated by super-resolution imaging of hMSCs on substrates of different stiffnesses, demonstrating the versatility of our approach in capturing chromatin organization changes. Our in silico experiments demonstrated how microenvironment-induced changes in chromatin modifier levels can lead to genome structure rearrangement and cell phenotype changes and revealed that epigenetic memory is predominantly encoded in regions flanking domain boundaries. The extent to which these flanking regions are affected by epigenetic reactions is variable and increases with the magnitude of the shifts in the magnitude of the extracellular stimulus. Ultimately, our hybrid experimental simulation approach elucidates the pivotal role of domain boundaries in maintaining cellular identity.

In summary, our chromatin polymer model yields two key findings, which we distill in our schematic (Fig. 8):

1. Chromatin domain formation and scaling emerge from the interplay between passive diffusion and active epigenetic reactions, as validated by super-resolution imaging of A375 melanoma cells. Furthermore, we revealed that chromatin domains exhibit scalable responses to substrate stiffness, adhering to the same scaling laws.

2. Changes in epigenetic reactions primarily impact the epigenetic landscape at domain boundaries, driving transcriptional state changes, as validated by Hi-C sequencing and RNA-seq. As a result, domain boundaries can also serve as critical control regions for the formation of microenvironment-driven epigenetic memory.

Our study provides a comprehensive understanding of chromatin dynamics and epigenetic regulation, underscoring the pivotal role of domain boundaries in shaping cellular identity. By demonstrating the applicability of our model to various scenarios where histone read-write mechanisms govern epigenetic regulation, we reveal the far-reaching implications of our findings. Notably, our analysis of Y-27 treatment and tendinosis highlights the model's utility in understanding biochemical treatment and disease progression. Moreover, the importance of histone epigenetic alterations extends to various diseases, including cancer onset and metastasis[106], Alzheimer's disease[107], and cell fate decisions[108]. Crucially, our work has the potential to reveal the central role of histone epigenetic modifiers, which can pave the way for the discovery of pharmacological interventions against cancer[109] and diabetes[110], among other disorders.

The formation of nanoscale chromatin domains creates rich heterochromatin–euchromatin interface at boundaries, which are energetically more susceptible to changes due to epigenetic enzyme fluctuations than other loci. We acknowledge that prior studies[44,53,111] have described spreading-driven boundary enrichment mechanisms, where spreading originates from a specific genomic locus and is shaped by the prescribed dynamics of epigenetic modifiers and other

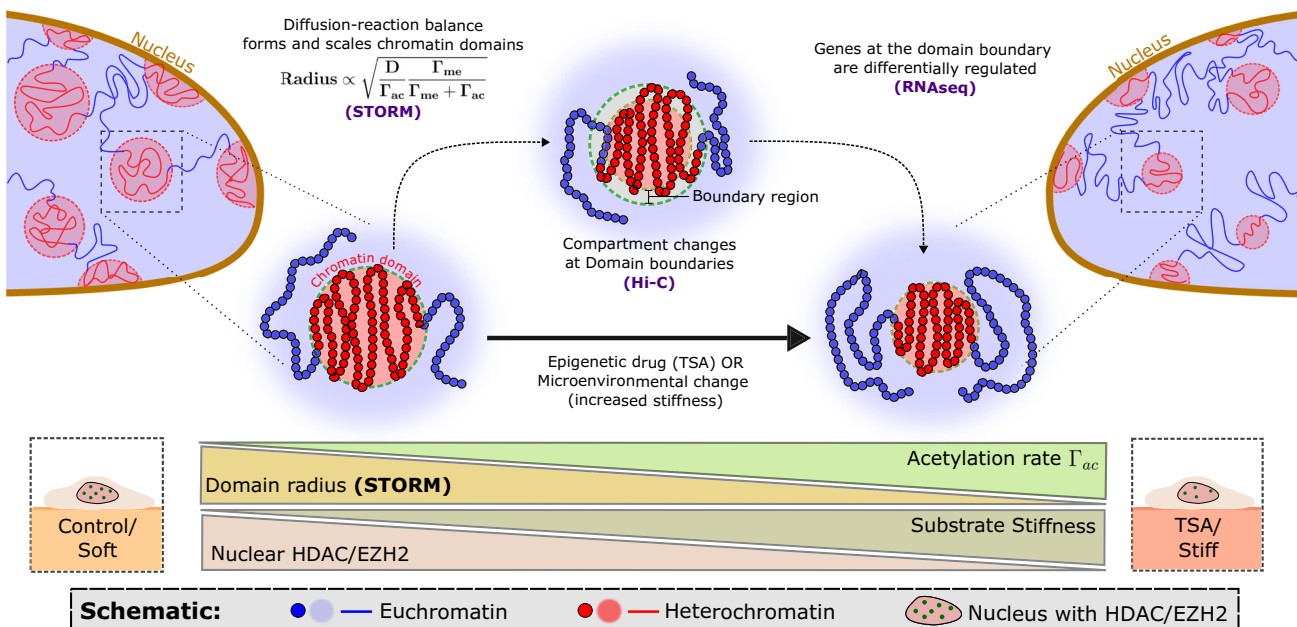

**Fig. 8 | Polymer model bridges STORM, Hi-C and RNA-seq to reveal the importance of domain driven chromatin organization and domain boundaries in genome regulation.** Our model shows that heterochromatin-rich chromatin domains form through a balance between diffusion and epigenetic reactions. In addition, a scaling relation between the domain size distributions and the epigenetic reaction rates is established, which captures the evolution of the chromatin organization with changing histone epigenetic remodeler concentrations. These scaling relations are validated through STORM imaging of A375 cells with TSA treatment and hMSCs with substrate stiffness change and GSK treatment. The model also predicts that epigenetic changes occur at the domain boundaries prominently. These predictions are confirmed by Hi-C sequencing with a high accuracy. The epigenetic changes lead to transcriptional changes close to the domain boundaries, especially affecting genes which are important for the cell line. In the case of A375 melanoma cells, we showed EMT-specific genes around the domain boundaries being affected. Lastly, we show that epigenetic memory can also store around the domain boundaries.

major players. Even though such mechanisms predict changes at the domain boundaries, our work departs from these earlier approaches in two key aspects:

1. Unlike previous models, which primarily focus on the spreading of epigenetic marks, our study emphasizes the spatial relationship between boundary domains and chromatin clutch domains as observed through STORM. Our findings demonstrate a direct relation between domain boundaries and the physical organization of chromatin clutch domains, a phenomenon we observe experimentally and validate through modeling. This aspect introduces an intriguing layer of biological relevance, connecting domain dynamics with the underlying chromatin architecture, which has not been explored in prior models.

2. Our model employs a deconstructed physical framework, where the out-of-equilibrium kinetics is governed solely by reaction rates, which break detailed balance. In contrast, previous studies rely on both active writer and active eraser mechanisms operating in tandem, making their models inherently more complex and departing it from the proposed mechanism in the involved timescales as well. This distinction allows our model to be more straightforward in its physical interpretation while still offering robust biological insights.

Our approach presents a mechanistic perspective by isolating the contributions of reaction-driven kinetics, providing clarity on how these mechanisms influence heterochromatic domain formation. By offering a physically and biologically interpretable framework, we believe our study complements existing literature and offers a rigorous mechanistic understanding of boundary positioning and dynamics.

Recent studies using dCas9-KRAB constructs support this notion[112], highlighting the dynamic nature of these boundaries. The recruitment of the RNAPII transcriptional machinery to these interfaces[102] further underscores its critical role in regulating the cell's epigenetic landscape. We propose that domain boundaries can serve as key chemo-mechanosensors, translating external cues into epigenetic changes that determine single-cell fate. This cell-invariant mechanism can enable cell line- and cell history-specific responses, which integrate with established signaling pathways to orchestrate transcriptional programs. For example, substrate stiffness-mediated activation of mechanosensitive pathways such as YAP/TAZ in mesenchymal cells[113] are well established, but our model demonstrates that a cell's microenvironmental history, encoded in its epigenetic and spatial configuration, can influence its response to such stimuli, yielding diverse outcomes. Our findings underscore the potential role of epigenetic landscape-driven domain boundary positioning in shaping cell-specific transcriptional programs.

### Limitations of the model and future directions
While our model has proposed key biophysical mechanisms governing chromatin organization and gene regulation, future refinements can address current limitations. In the current setup, we have considered a copolymer simplification, which is a major hinderance in the capabilities of the model. Hence, incorporating multistate chromatin models that account for different epigenetic flavors (such as ChromHMM[61] or MEGABASE[34]), chromatin–lamina interactions[114], and sequence-dependent epigenetic marking[115] will increase the predictive power. In addition, the influence of other molecular machinery on chromatin organization has not been considered. For instance, RNA Polymerase II-driven supercoiling[116] is known to play a crucial role in

chromatin organization. Similarly, condensing proteins such as HP1[56], transcription factor binding[117], and cohesin-driven loop extrusion[118] significantly impact the spatial positioning of genomic loci. These factors are essential in shaping the 3D chromatin organization, which is central to the epigenetic and transcriptional shifts predicted by the model.

Lastly, the model also has spatial as well as temporal limitations. All the simulations we have performed in this study are over a single chromosome, while it is well known that regions in between chromosomes can play the role of transcription hubs and, hence are central to determining expression. Temporally, we capture and predict changes within a single cell cycle. Integrating replication dynamics[119] and epigenetic spreading mechanisms[120] will reveal long-term chromatin architecture changes, which are crucial for understanding cancer metastasis and development. This will require us to incorporate the role of central proteins, such as condensin motors which drive sister chromatid formation and eventual cellular division[121]. HP1 proteins have also been shown to be essential in determining the epigenetic stability of histone modifications across cell cycles[122]. Lastly, exploring the interplay between chromatin organization and nuclear signaling pathways will provide a more comprehensive understanding of cell fate determination. By building upon our foundation, future studies can investigate the role of chromatin organization in disease mechanisms, ultimately informing therapeutic strategies. The integration of experimental and theoretical approaches will be essential for achieving a holistic understanding of chromatin organization and its role in governing cell fate.

## Methods

### M1 Data-driven construction of the polymer model

To model chromatin organization, we represent chromatin as a self-avoiding beads-on-a-string polymer, where each bead corresponds to a 10-kb genomic segment. Each bead is assigned an epigenetic state—euchromatin-like (active) or heterochromatin-like (repressed)—based on experimental annotations (Fig. 1a). The polymer configuration follows a block copolymer approach, where chromatin compartments emerge from preferential intra-type interactions. To ensure both biophysical relevance and computational efficiency, we adopt a two-pronged approach to chromatin polymer initialization. First, a randomly initialized polymer (3000 beads, 60% repressed) is used to investigate fundamental biophysical principles governing chromatin compaction. Second, a data-driven polymer initialization incorporates experimentally derived chromatin states. For human mesenchymal stem cells (hMSCs), we use Broad ChromHMM data to classify beads into euchromatin or heterochromatin compartments of chromosomes 19–22. For A375 melanoma cells (chr18-21), chromatin states are assigned based on Hi-C contact maps, ensuring that the polymer reflects known A/B compartmentalization. These complementary approaches allow us to validate the model's robustness across data-agnostic and data-driven configurations.

The interactions between chromatin segments are governed by a Lennard-Jones potential, which captures the influence of bridging proteins (e.g., HP1, PRC1) and nonspecific chromatin affinities (Fig. 1b). Attractive interactions are restricted to intra-type chromatin segments to enforce epigenetic compartmentalization. The interaction strength is parameterized as $\epsilon\_HH$ (heterochromatin-heterochromatin) and $\epsilon\_EE$ (euchromatin-euchromatin), with the constraint $\epsilon\_HH > \epsilon\_EE$, ensuring greater heterochromatin compaction while allowing euchromatin to remain more dispersed. Excluded volume interactions are enforced between heterochromatin and euchromatin segments, limiting nonspecific interactions. To maintain polymer integrity, adjacent beads are connected by a finite extensible nonlinear elastic (FENE) potential, which prevents overextension while preserving polymer flexibility. The polymer is thermally equilibrated via Langevin dynamics, incorporating stochastic damping forces to simulate the

viscous nuclear environment. This ensures that the simulated chromatin structure adheres to biologically relevant polymer behavior.

To assess the validity of the chromatin polymer model, we compare simulation outputs to experimentally observed chromatin organization patterns. The model successfully recapitulates:

- The spatial distance-genomic distance relationship observed in chromosome conformation capture experiments (Hi-C) (Fig. 1d).
- The characteristic Hi-C checkerboard pattern of A/B compartmentalization, which emerges naturally from the interaction constraints imposed by the polymer model (Fig. 1e).

While cohesin-dependent loop extrusion plays a key role in chromatin topology, it is not explicitly modeled in this study. At the current level of coarse-graining, we do not expect significant deviations in chromatin organization due to the absence of loop extrusion. However, our model retains the flexibility to incorporate additional architectural factors (e.g., CTCF binding, loop extrusion) in future extensions (details in Supplementary Information 1.1).

### M2 Diffusion and Epigenetic reactions

We explain the proposed dynamics and their underlying implementations in more detail as follows:

**Brownian diffusion.** The spatial diffusion of chromatin within the nucleoplasm is captured by employing Brownian dynamics to simulate the diffusion of chromatin beads (Fig. 1f). Considering a viscosity of 150 cp[41,123] and a bead size of 65 nm, we estimate a Brownian time step $\tau_{Br}$ of 0.3 s (calculations provided in Supplementary Information 1.3). To simulate, we choose an integration time step $\Delta t$ of $0.01\tau_{Br}$. This choice allows us to simulate ~1 h of real-time behavior for every $1 \times 10^6$ integration step. We expand on the choice of our parameter selection and its effects in obtaining the effective viscosity and diffusivity in Supplementary Information 1.3.

**Diffusion of epigenetic marks.** Given that interactions between heterochromatic regions are more favorable than those between heterochromatin and euchromatin, it is unfavorable for isolated heterochromatin segments to be in a euchromatin-rich environment. When the overall ratio of the hetero and euchromatic beads remains constant, the system tends toward a state with lower energy, leading to the phase separation of euchromatin and heterochromatin. To illustrate this further, we designed a schematic, as shown in Fig. 1g, where the fully phase-separated configuration is more favorable than the other configurations because of unfavorable interactions between chromatin segments. An energetically driven exchange between neighboring regions occurs as we transition from the top to the bottom configuration. This exchange lays the foundation for the diffusion of epigenetic marks. We explain the motivation for including this diffusive mechanism in more detail in Supplementary Information 1.7. We employ a Monte Carlo-based epigenetic labeling exchange algorithm to account for this phenomenon. Since the epigenetically conserved process should maintain a thermodynamic balance and lead to redistribution of only the epigenetic marks, we take inspiration from Kawasaki dynamics, which are used to study spin-conserving Ising systems[124] (Fig. 1g). The steps of our exchange algorithm are as follows:

1. Choose a random chromatin bead i and a neighboring bead j (within $1.8\sigma$).
2. The energy change $\Delta E$ associated with swapping the epigenetic marks of beads i and j is calculated. The energy change is given by the difference in energy before and after the epigenetic labeling swap.
3. If $\Delta E \leq 0$, accept the swap and update the chromatin labeling configuration by exchanging the epigenetic labels at sites i and j.
4. If $\Delta E > 0$, accept the swap with a probability $P = \exp(-\Delta E/kT_E)$, where k is the Boltzmann constant and $T_E$ is the temperature

associated with this exchange. This probability, known as the metropolis criterion, ensures that the system tends to lower its energy over time in a canonical ensemble setting.

To ascertain that such dynamics will not give rise to out-of-equilibrium components and lead the system to a thermodynamically favored state, we show that it follows a detailed balance in Supplementary Information 1.4. We note that this diffusion is performed in 3D and not along the polymer strand, as shown explicitly in Supplementary Fig. 4.

**Epigenetic reactions.** To account for changes in the concentration of nuclear epigenetic remodelers, which can be driven by the microenvironment[23,25,26,29], we employ epigenetic reactions. Epigenetic reactions exhibit active dynamics, act in a nonconservative manner, and can thus change the net heterochromatin-to-euchromatin (red-to-blue) ratio in the system. Heterochromatin can be converted to euchromatin through the activity of histone demethylases followed by the action of histone acetyltransferases. Here, we quantified the conversion of heterochromatin to euchromatin because of the combined effect of these reactions on the acetylation rate ($\Gamma_{ac}$). Similarly, we quantified the conversion of euchromatin to heterochromatin through the methylation rate ($\Gamma_{me}$). These reactions are modeled as stochastic processes in which a random bead is chosen and converted to heterochromatin or euchromatin bead with rates equal to $\Gamma_{me}$ (methylation) or $\Gamma_{ac}$ (acetylation), respectively (Fig. 1h). When a reaction move is executed, the epigenetic marking of a chosen bead is changed in a probabilistic manner, i.e., the bead changes to blue with probability $p_{ac}(\Gamma_{ac})$ and to red with probability $p_{me}(\Gamma_{me})$ such that $p_{ac}(\Gamma_{ac}) + p_{me}(\Gamma_{me}) = 1$. We show that epigenetic reactions break the detailed balance and introduce out-of-equilibrium dynamics in Supplementary Information 1.4.

The Brownian diffusion timescale ($\tau_{Br}$) is taken as the reference timescale for the system. In addition, we execute one epigenetic diffusion move with every Brownian step, which gives us an effective diffusivity $\sim 1 \mu m^2/s$[125,126] (details in Supplementary Information 1.3). To execute the reactions, we execute one epigenetic reaction step for every $10^3$ simulation steps. Since each simulation step corresponds to $10^{-4}$ s, the reaction rate corresponds to a timescale of $\sim 0.1 \, s^{-1}$, which is in the range of experimentally observed histone remodeler activity[127]. To perform production runs and simulate real-time chromatin organization changes, we first constructed a polymer from experimentally observed sequencing data, as detailed in section Chromatin polymer model follows experimentally observed length scales and 119 incorporates diffusion and epigenetic reaction dynamics. Thereafter, we subject it to diffusion and reaction dynamics.

## M3 Preparation of cells and TSA treatment
A-375 (CRL-1619) cells were procured from ATCC. A-375 cells were isolated from a 54-year-old female patient with malignant melanoma. They possess an epithelial morphology. The cells were resuspended in basal growth medium (DMEM and 10% fetal bovine serum (FBS)). Experiments were performed on cells at greater than 3 passages or fewer than 8 passages. Splitting was performed only after a confluence of more than 80% was achieved.

We used tricostatin A (TSA) as an HDAC inhibitor (Fisher Scientific 14-061). For TSA treatment, the cells were grown to > 70% confluence for more than 24 hours before the addition of 500 nM TSA along with basal media. These were returned to the incubator and cultured for another 2 h. Thereafter, they were washed four times with PBS for 5 min each and then fixed with 4% paraformaldehyde (PFA) in PBS for 10 min (imaging) or 1% formaldehyde in 1 × HBSS (genomic assays).

For immunostaining and confocal imaging, after incubation, primary antibody (H3K9ac) diluted in antibody dilution buffer (4% BSA, 0.25% Triton in PBS) was added, and the samples were incubated overnight at 4 °C. After primary antibody incubation, the cells were washed 3 times with PBS for 5 min each. The cells were then incubated with secondary antibodies (either Alexa Fluor 594 or Alexa Fluor 488) per the manufacturer's directions for 30 minutes at room temperature. After secondary antibody incubation, the cells were washed 3 times with PBS and sealed with mounting media supplemented with DAPI. Slides were treated with mounting media for 24 hours before imaging. The cells were imaged via a Leica Sp8 confocal microscope equped with a 63x oil immersion objective.

## M4 STORM imaging and analysis
To perform STORM imaging of the fixed sample, the cells were incubated in a blocking buffer containing 10% wt/vol BSA (Sigma) in PBS for 1 h. The samples were then incubated overnight with rabbit anti-H2B (1:100; Proteintech, 15857-1-AP) at 4 °C. Thereafter, the samples were repeatedly washed in PBS, and secondary antibodies (Alexa Fluor 647) were added for imaging. The images were taken on a commercially available ONI (Nanoimager S) STORM microscope system. To ensure optimal photo-switching of Alexa Fluor 647, the imaging buffer followed standard guidelines[128] and consisted of 10 mM cysteamine MEA in Glox solution: 0.5 mg ml-1 1-glucose oxidase, 40 mg ml-1 1-catalase and 10% glucose in PBS. A 640 nm laser was used at a setting of 40% to excite the reported dye (Alex Fluor 647), and a gradual increase in the 405 nm laser was used to reactivate Alexa Fluor 647 in an activator dye to maintain a constant intensity of active fluorophores. An exposure time of 15 ms and 30 k frames was used. Downstream analysis was performed with the help of a custom code in MATLAB (Supplementary Information 3.1), which we showed in a previous publication[11].

## M5 Hi-C preparation
Biological replicates of Hi-C were performed on two independent samples of A375 cells, both untreated and treated with 0.5 μM TSA for 2 hr. The first biological replicate was processed with the protocol described previously[129]. Briefly, 10 million cells were crosslinked with 1% formaldehyde for 10 min. Crosslinked cells were then suspended in lysis buffer to permeabilize the cell membrane and homogenized. Chromatin was then digested in the nucleus overnight using the DpnII restriction enzyme. The digested ends were filled with biotin-dATP, and the blunt ends of the interacting fragments were ligated together. The DNA was then purified via phenol–chloroform extraction. For library preparation, the NEBNext Ultra II DNA Library prep kit (NEB) was used for libraries with sizes ranging from 200–400 bp. End Prep, Adapter Ligation, and PCR amplification reactions were carried out on bead-bound DNA libraries. The second biological replicate was processed via the Arima-HiC + Kit from Arima Genomics following the protocol for Mammalian Cell Lines (A160134 v01), and the libraries were prepared via the Arima recommendations for the NEBNext Ultra II DNA library prep kit (protocol version A16041v01).

Sequencing was performed on a HiSeq 3000 platform with 150 bp paired-end reads. Sequencing reads were mapped to the human genome (hg19), filtered, binned, and iteratively corrected as previously described[130] via the HiCPro pipeline [https://github.com/nservant/HiC-Pro]. Spatial compartmentalization (A/B compartment assignment) was calculated via principal component analysis with 40 kb binned data.

## M6 RNA-seq preparation
RNA was extracted from TSA-treated and untreated A375 cells via the Qiagen RNeasy Plus Mini Kit (Cat. No. 74134). The cells were lysed, homogenized, and spun down in gDNA eliminator columns to remove any genomic DNA. All the samples were washed with ethanol, and the total RNA was eluted. The quality and quantity of the RNA were quantified via an Agilent Bioanalyzer Nano RNA Kit. All the libraries used were characterized by an RNA integrity number (RIN) between 9 and 10. The RNA was sent to the Oklahoma Medical Research

Foundation for library preparation (using an rRNA depletion approach) and sequencing.

The quality of the reads was checked via FastQC, and adapter sequences were trimmed via BBTools [https://github.com/kbaseapps/BBTools]. In addition, quality trimming of the reads was performed, and any reads with a quality score lower than 28 were discarded. The reads were then aligned via STAR alignment [https://github.com/alexdobin/STAR]. The aligned reads were then sorted by genomic position, and feature counts were performed via HTSeq 0.11.1 [https://github.com/simon-anders/htseq]. The differential expression of genes was determined via DESeq2[131].

**M7 Statistics & reproducibility**

The study design was developed to ensure robust and reproducible results. Sample sizes were determined based on prior literature and feasibility considerations, rather than through a formal statistical power analysis. No data were excluded from the analyses. The experiments were not randomized, and investigators were not blinded to allocation during experiments or outcome assessment. Data processing and statistical analyses were conducted using Python and R, with statistical significance assessed using t-tests.

**Reporting summary**

Further information on research design is available in the Nature Portfolio Reporting Summary linked to this article.

## Data availability

The sequencing raw files are available on the GEO dataset with the accession number: GSE275755. Additional data supporting the findings of this study are available from the corresponding author upon request. Requests will be catered to promptly and appropriate time will be provided to facilitate data transfer. Source data are provided in this paper.

## Code availability

The code used for measurement of sizes of heterochromatin domain obtained from STORM imaging is freely available through GitHub[132]. The simulations have been done in LAMMPS[133] (stable release Aug 2023) with custom fixes written while the analyses is done in python with codes available on GitHub[134]. The Hi-C analyses is performed using cooltools[135] and RNA-seq using DESeq2[131] and clusterprofiler[69]. All the figures were assembled using Inkscape[136].

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

## Acknowledgements

The authors thank Aayush Kant, Monika Dhankhar, and Zixian Guo for the multiple useful conversations and suggestions and for sharing their work on the STORM analyses code. We thank Christopher Playter and Priyojit Das (UTK) for assisting with the Hi-C experiments and analyses. This work was supported by NIH Award U54CA261694 (V.B.S.); NCI Awards R01CA232256 (V.B.S.); NSF CEMB Grant CMMI-154857 (V.B.S.); NSF Grants MRSEC/DMR-1720530 and DMS-1953572 (V.B.S.); and NIBIB Awards R01EB017753 and R01EB030876 (V.B.S.). This work was supported in part by the National Institutes of Health [NIGMS grant R35GM133557 to R.P.M]. We are deeply grateful to the reviewers for their thoughtful and constructive feedback, which significantly strengthened this work.

## Author contributions

V.V. and V.B.S. conceived the project. V.V. led the development of the model. V.V., R.B., and L.S. conducted the computational and numerical analyses for the polymer model. R.G. and R.P.M. performed the Hi-C and RNA-seq experiments, with sequencing analyses carried out by V.V., R.G., and R.P.M. V.V., J.T., and M.L. conducted the super-resolution imaging and analyses. V.V. and V.B.S. led the drafting of the manuscript, with R.B., J.T., M.L., and R.P.M. contributing to revisions and edits.

## Competing interests

The authors declare no competing interests.
