## [Peer Review file · Nature Communications]

Polymer Model Integrates Imaging and Sequencing to Reveal How Nanoscale Heterochromatin Domains Influence Gene Expression

Corresponding Author: Professor Vivek Shenoy

Version 0:

Reviewer comments:

Reviewer #1

(Remarks to the Author)

Review of Vinayak et al.

This manuscript develops features in bead-spring polymer models that enable an understanding of the physical principles that underlie partitioning of active and inactive chromatin. The major feature they introduce is a function to differentiate beads into two categories and an algorithm for switching. Remarkably, this functionality introduces sufficient heterogeneity into the model to help us understand basic features in live cells. The manuscript is well-written and clearly described.

Specific comments.

The authors cite a viscosity of 150cp for their beads. The references are from a Rippe paper (DOI: 10.1038/ncomms5494) or from a modeling paper. The viscosity they use should not be the fluid phase of the nucleus, rather the beads in a polymer experience a much higher viscosity. RB Nicklas estimated a viscosity of 1P for a chromosome (Nicklas RB (1983)

Measurements of the force produced by the mitotic spindle in anaphase. J Cell Biol 97:542–548) and others estimate significantly higher for chromatin in living cells (141P www.pnas.org/cgi/doi/10.1073/pnas.0812723106). I would ask that the reviewers cite more appropriate values for a bead in a chain and justify their use.

They discuss an effective diffusivity of 1 $\mu\text{m}^2/\text{sec}$ (line 234). I'm not sure what they are referring to. Chromosomes have a diffusivity of $5 \times 10^{-12} \text{ cm}^2/\text{sec}$, or $5 \times 10^{-4} \text{ cm}^2/\text{sec}$. These beads are in a polymer chain and moving considerably slower than a free bead. The main impact these numbers have on the simulation is the time scale, however it is very important as they are trying to make a case for the timescale and consistency with remodeler activity (Line 233).

The authors did a commendable job showing how the model interacts with various experimental measurements. However, they overstate their case stating that "reactions and diffusion compete to give rise to chromatin domain" "a result that has not been previously reported". A variety of studies introduce features that reduce the homogeneity of a simple bead spring model. Some of these are densities of GC-content, CtCF sites doi: 10.1038/s41598-017-02923-6; density and distribution of cross-linkers doi: 10.1093/nar/gkx741; different chromatin states open and dumbbell <https://doi.org/10.7554/eLife.60312>).

What these authors have implemented is a bead exchange algorithm that is energetically favorable. Based on prior studies, there is every expectation that heterogeneity in the system will arise and it is no surprise they can tune the energetics to fit experimental data.

I found the manuscript to be overall interesting and informative, and certainly contributes to the large body of polymer modeling that is serving the field well in providing physical frameworks for understanding the complexities of chromosome structure and function.

(Remarks on code availability)

Reviewer #2

(Remarks to the Author)

In this work the authors introduced a new polymer model to simulate chromatin compartmentalization – a fundamental architectural feature in 3D genome organization. Unlike previous models that mostly consider only the passive compartmental interactions among regions with the same compartmental identities, this new model creatively includes

active reaction that allow regions within compartments to change compartment identity, as well as diffusion of compartment identities. This model is refreshing and establishes a new framework for the theoretical understanding of compartment organization maintenance, reorganization, and epigenetic memory. In addition, the work included a series of experimental tests of the model, demonstrating the validity of the model. This work for sure deserves publication in Nature Communications, and I have only a few minor points to help strengthen the paper:

1. The new model with the reaction-diffusion mechanism nicely explains why compartment domains do not reach full phase separation but maintain as many small domains. A conventional model that does not depend on this reaction-diffusion mechanism would be: entropic penalty prevents the polymer from reaching full phase separation with the minimum enthalpy. I guess if the A-A and B-B interaction strengths are tuned lower and lower, at some point, even without the reaction-diffusion mechanism, the chromatin will not reach full phase separation but maintain as many small domains, because this is entropically favored. This conventional model is potentially consistent with the experimental results as well, as the acetylation or methylation reactions could be modeled as changing the A-A and B-B interaction strengths. Could the authors clarify: Is the new model better than the conventional model, or is it an alternative to the conventional model? In either case, the new model should be published as it represents a conceptually different framework that is inspiring to the field.
2. An ideal experiment that can fully test the validity/contribution of the new model vs. the conventional model to real compartment organization would be: Treat the cells with a cocktail of inhibitors that stops all acetylation and methylation reactions, so that the epigenetic identities along chromatin are "fixed". Then if the new model is correct, the A-B compartments in the cells would reach full phase separation without the reactions. If the conventional model is correct, the A-B compartments in the cells would not reach full phase separation since the entropic penalty is still there. I'm not sure if this experiment is doable but the authors could discuss and suggest this possibility for a future work, as this will generate fundamental insights regarding which model is correct (or how much each model contributes to reality, since they are not completely exclusive to each other).
3. In terms of the diffusion, my understanding is the authors allow only diffusion in 1D along the DNA, not in 3D when two regions are in contact. It would be nice if the authors can explicitly make this clear that 3D diffusion is not included in this model.
4. The authors showed in simulation that upon changes of reaction rate, compartment identity changes are enriched at domain boundaries. This phenomenon needs a better explanation. Based on the model description, random beads are chosen to be converted to heterochromatin or euchromatin beads with defined rates. How come the chromatin regions at domain boundaries are more often altered? My guess is: after a random bead is converted, the new bead identity (in a domain of opposite identity) can diffuse, and when it diffuses to the boundary, it is "fixed" there due to the favorable energy. As a result, after simulation of a time period, more alterations are "fixed" at the boundaries. Is this guess correct?
5. Line 697: After "Heo et al" a citation number is missing.
6. Line 764-765: "When undergoing extracellular:" A word is missing here, likely "stimulation".
7. Line 783-785: "Therefore, smaller domains can act as domain boundaries themselves and give rise to lasting memory characteristics". I thought the results above just showed that smaller domains do NOT give rise to lasting memory and can disappear upon perturbation, and upon removal of perturbation do not reappear at the same region but at nearby region. The bigger domains give rise to lasting memory and at least partially recover.

(Remarks on code availability)

Reviewer #3

(Remarks to the Author)

The work by Vinayak et al. explores the role of epigenetic reaction rates on heterochromatin domains employing data informed polymer model and experiments. They pose active epigenetic reactions as an opposing force to diffusing epigenetic domains. The authors have also investigated experimental perturbations and their effect on chromatin organization and gene expression. The first part of the work sets-up the model and the later part focuses on making perturbations to the epigenetic reactions (altering rates) to study the effects. This is an interesting and relevant work. However, several concerns (see below) need to be addressed before being considered for publication.

Major Comments

1) The concept of diffusion of epigenetic marks: At the molecular level, epigenetic marks being covalent modifications on histone proteins or DNA, established by enzymatic reactions (active). While as purely mathematical model its interesting, but since being contextualized to chromatin, I find it hard to understand the basis (biological context or origin) for having "diffusion of epigenetic marks" or the use of Kawasaki dynamics here. For instance, several experiments have shown that heterochromatin domains to decay completely when enzymes are mutated (Ex: Audergon et al. Science 2015, Ragnathan et al. Science 2015). The redistribution/diffusion of epigenetic marks is a puzzling choice.

2) By construction the model is a competition between diffusion and active reactions, this makes all perturbation on rates affect the domain boundaries as nicely illustrated in Figure 2. Alternatively, previous models based on spreading of epigenetic marks from a nucleation site present a mechanism where domain boundaries are affected on changing rates of 'epigenetic remodelers' as well (Ancona et al. PRE 2022, citation [45,54], for instance, figure 2 in citation [45]). Discussing and contrasting with alternate possibilities/mechanisms previously reported is essential.

Detailed Comments

Figure1:

- 1) For better reproducibility, include the parameters for figures d and e in the figure description ($\epsilon_{EE} = 0.3$ and $\epsilon_{HH} = 1$). Why was a 'hit-and-try' method used instead of exploring the parameter space (at least around the hit) showing the fit-score. This would also inform on how sensitive the system is to the parameter.
- 2) Figure S2, Random-walk initialization potentially gives rise to knots in the system. Is there a change in the kinetics if an alternate knot-free initialization is used?
- 3) Timescales: Histone turnover rate (~ per hour) in Alabert, C et al. Genes and development 2015, hours to days in citation [69], Aimee M Deaton et al. eLife 2016 (faster turnover rate) and HMT based reaction rates (min^{-1}) in Newar et al PLoS CB 2022 seems to show a lot of variability depending on cell-type/stage. Since there is no explicit 3D organization dependent spreading, how would the ratio of epigenetic reaction rate and polymer dynamics (currently one reaction per 10^3 simulation step) affect the system? Maybe in the supplementary, include a figure to illustrate the effect.

Figure 2:

- 4) Mention the reaction rates used in the figure description. Why does it require the use of DBSCAN to compute Rg, isn't the spatial x,y,z coordinates of the monomers known?
- 5) Since the time mapping has been already done, it would be more informative if the x-axis of figure a is in real time, the conversion maybe be mentioned in the description.
- 6) Irrespective of the initial distribution, based on the ratio of reaction rates (me/ac), can we predict the number of distinct, persistent separated domains or is it stochastic? While the model can sustain distinct, separated domains, won't the domains be of the same size at steady state?

Figure 3:

- 7) Quantify the fraction of monomers that retain the same state between $t=0$ and $t=2$ hours, mention it in the description for figure a and d.
- 8) Show the pairwise distance map comparison for the initial and final state for the case shown in figure 3a, it would help the comparison. Also, in the supplementary plot a difference map to quantify and highlight the difference between the timepoints. Mention the energy parameters in the main text.
- 9) Extending question 6, doesn't figure c show a peaked distribution upon the ratio of rates being slightly skewed, implying the model predicts most of the domains to have uniform size? Does the mean domain size remain the same after the perturbation or what is the difference?

Figure 4:

- 10) The section heading "STORM imaging confirmed chromatin domain size scaling in agreement with model prediction" is misleading because it doesn't confirm the scaling argument presented in equation 1.
- 11) TSA treatment shows a decrease in heterochromatin, upon increasing acetylation impairing heterochromatin is expected (Zofall M. et al. 2022 NSMB, Sahu R K et al. Mol Cell 2024, among many other studies) and is that a key prediction?
- 12) In the simulation (figure c), what is the increase in acetylation rate and how was this value chosen? Isn't there an acetylation rate change that would give closer to 8% change in mean? Wouldn't the behavior be the same if methylation rate is reduced? So, can an alternate perturbation inhibiting methyltransferase be used to validate the model?
- 13) Is the take-away from the section "averaging over cell population leads to fewer epigenetic compartmental changes" that averaging or bulk data misses out ON cell-to-cell variability? Isn't that the basis of averaging, to see changes that are present more frequently in population?
- 14) Can the shift be attributed to large domains and small domains upon TSA treatment, while intermediate domains remain unaffected as the analysis in figure 3 would suggest? In figure c the shape of the distribution experimental TSA and simulated TSA seems quite different, why is that so?

Figure 5:

- 15) 30% of DE genes are within 300 kb of a boundary element, in the model/simulation does TSA treatment change that happens at the boundary (300kb) as opposed to more central?

Figure 6:

- 16) The claim that polymer model captures chromatin re-organization in response to changes in microenvironmental stiffness is misleading because the polymer model only acts through methylation/acetylation rate irrespective of the mechanism by which this change is brought about by (chemically or mechanically induced). Mechanistically the model doesn't link environmental stiffness change to organization rather plays only on acetylation/methylation rate as input.

Figure 7:

- 17) Epigenetic memory is generally used to refer to retaining epigenomic state post-replication (in the context of heritability). It would be interesting to see how incorporating replication (dilution of methylation/acetylation) would affect the behavior.
- 18) A remark on citation: Earlier works Sandholtz S H. et al. PNAS 2020 and citation [45] illustrate chromatin organization and limited enzymes allow stable epigenetic memory.

Discussion section:

- 19) The discussion section would benefit from a detailed section on the limitations of the model, possible perturbations the model does not capture.

(Remarks on code availability)

Version 1:

Reviewer comments:

Reviewer #1

(Remarks to the Author)

The authors have adequately addressed the reviewers concerns.

(Remarks on code availability)

Reviewer #2

(Remarks to the Author)

The authors have fully addressed my comments. I recommend the paper for publication. The only optional suggestion is I find the following response paragraph from the authors very satisfyingly addressed my No. 1 question. The authors can consider including this and Fig. R2 as a supplementary analysis in the supplementary materials.

"Consistency with experimentally observed length scales: It is important to note that the interaction parameters (ϵ_{HH} and ϵ_{EE}) are tuned against experimental data such that they follow the observed spatial distance to genomic distance parameter. Hence, while tuning the A-A and B-B interaction strengths to lower values (as suggested by the reviewer) may indeed maintain chromatin domain formation and prevent them from ripening due to entropic effects, this approach will introduce discrepancies between the observed chromatin organization length scales and the computationally produced length scales. We show the effect of reducing the interaction potentials such that we have lower A-A and B-B interactions in Fig. R2 (corresponding to Fig 1d). This clearly shows that such a model would not fit the observed length scales. In contrast, the potentials in our model are calibrated to replicate experimentally observed scaling relations, ensuring consistency with available data."

(Remarks on code availability)

Reviewer #3

(Remarks to the Author)

(Remarks on code availability)

We sincerely thank the reviewers for their thorough evaluation of our manuscript and for their valuable comments and suggestions. We have carefully addressed each point in a detailed, point-by-point response. The additional information and revisions made to the manuscript and supplementary materials comprehensively address the concerns raised. We believe these enhancements have strengthened the manuscript significantly.

(The reviewer comments are in plain text, our response in blue, and the edits made to the manuscript are highlighted in yellow.)

Reviewer #1:

1. The authors cite a viscosity of 150cp for their beads. The references are from a Rippe paper (DOI: 10.1038/ncomms5494) or from a modeling paper. The viscosity they use should not be the fluid phase of the nucleus, rather the beads in a polymer experience a much higher viscosity. RB Nicklas estimated a viscosity of 1P for a chromosome (Nicklas RB (1983) Measurements of the force produced by the mitotic spindle in anaphase. J Cell Biol 97:542–548) and others estimate significantly higher for chromatin in living cells (141P www.pnas.org/cgi/doi/10.1073/pnas.0812723106). I would ask that the reviewers cite more appropriate values for a bead in a chain and justify their use.

Response:

We thank the reviewer for highlighting this important parameter choice. In our non-dimensionalized MD framework, the chosen viscosity determines the timescales, making its selection crucial. We acknowledge that determining the appropriate viscosity for chromatin is complex, as chromatin behaves as a highly entangled polymer with associated proteins, and therefore does not follow the simple hydrodynamic drag relationships observed for smaller proteins diffusing in the nucleoplasm. As the reviewer notes, this has led to a wide range of reported viscosity values. For example, Nicklas (J Biol, 1983)[1] reports a viscosity of 1 P. On the other hand, Fisher et al. (PNAS, 2009)[2] report a much higher viscosity of 147 P, and they themselves note that "in vivo viscosity values

Figure R1: MSD calculations of the polymer showing different diffusive regimes. Top left: MSD calculated over 10 independent runs over the course of ~1hr (real time). Top right: A log-log plot with a linear fit exhibiting diffusion in the $\alpha = 0.49$ regime. The fit produces $D_\alpha = 7 \times 10^{-2} \mu\text{m}^2/\text{s}$. Bottom left: MSD for ~10 seconds. Bottom right: The plot with a linear fit exhibiting diffusion in the $\alpha = 0.98$ regime. The fit produces $D_\alpha = 4 \times 10^{-3} \mu\text{m}^2/\text{s}$.

As the reviewer notes, this has led to a wide range of reported viscosity values. For example, Nicklas (J Biol, 1983)[1] reports a viscosity of 1 P. On the other hand, Fisher et al. (PNAS, 2009)[2] report a much higher viscosity of 147 P, and they themselves note that "in vivo viscosity values

obtained from our entropic relaxation experiments are quite high" when compared with other published data, placing their reported values at the upper end of the spectrum.

To address the reviewer's concern regarding the viscosity assigned to beads in our polymer model, we provide the following rationale: Considering the conservation of total chromatin and water (nucleoplasm) volume, the movement of chromatin in one direction necessitates the displacement of water in the opposite direction. This reciprocal movement implies that the viscosity experienced by chromatin monomers reflects the effective viscosity of the combined chromatin-water system, rather than that of chromatin alone. By utilizing the nucleoplasmic viscosity as our input parameter, we aim to accurately represent the hydrodynamic drag on diffusing chromatin segments due to the nucleoplasmic fluid, while internal polymer dynamics, including cross-linking interactions and neighboring bonds, are explicitly modeled through the polymer's interaction potentials. This dual approach facilitates a more physiologically relevant depiction of chromatin motion within the nuclear milieu, accounting for both the frictional drag from the fluid and the internal resistance from the polymer network.

Additionally, the transport of water relative to chromatin is governed by two key physical parameters: viscosity and permeability. Together, these factors influence the overall diffusion behavior of chromatin. Permeability determines the extent to which nucleoplasmic water can traverse the chromatin network, while viscosity dictates the resistance to flow. The interplay between these parameters is mathematically linked to the effective diffusion coefficient, as observed in studies of particle motion in porous media and polymer gels[3]. The dynamic interaction of viscosity and permeability significantly affects the anomalous diffusion of chromatin, often manifesting as a transition between different regimes over different timescales. By explicitly modeling chromatin-chromatin interactions and hydrodynamic drag, we effectively account for these contributions without modifying the input viscosity. This methodology parallels models of intracellular transport, where effective diffusion coefficients describe movement in crowded, viscoelastic environments. Our subsequent analysis of the mean squared displacement (MSD) for chromatin supports this interpretation, revealing sub-diffusive dynamics at longer timescales and Brownian-like motion at shorter timescales, consistent with prior experimental observations[4].

To quantify the effective viscosity, we computed the mean square displacement (MSD) of the chromatin polymer in our simulations and found that it exhibits two different diffusion regimes, one for times less than ~10 seconds and another on longer timescales, similar to that found in previous studies[4]. To quantify the diffusion relations, we fitted the two parts to the anomalous diffusion equation: $\langle r^2(MSD) \rangle = 2D_\alpha t^\alpha$. Using this, we obtained values of $\alpha \approx 0.5$ and $D_\alpha \approx 0.07 \mu m^2/s$ (Fig. R1) for the longer timescales, which corresponds to sub-diffusive motion. In this regime there will be an effective viscosity of $\eta \approx 30P$. For shorter timescales (<10sec), we obtain $\alpha \approx 1$ and $D_\alpha \approx 4 \times 10^{-3} \mu m^2/s$, which corresponds Brownian dynamics and has an effective viscosity of $\eta \approx 10^3P$. Both these values fall well within the range of observed viscosity for the chromatin polymer and other large droplets within the nucleus, which range from 1P[1]-1000P[5].

In response to the reviewer's suggestion, we have referenced to this discussion in the Methods section and SI1.3 to include this discussion which provides a detailed rationale for

our choice of viscosity, the associated diffusion. We believe these additions address the reviewer's concern while also enhancing the overall impact and clarity of the manuscript.

2. They discuss an effective diffusivity of $1 \mu\text{m}^2/\text{sec}$ (line 234). I'm not sure what they are referring to. Chromosomes have a diffusivity of $5 \times 10^{-12} \text{ cm}^2/\text{sec}$, or $5 \times 10^{-4} \text{ cm}^2/\text{sec}$. These beads are in a polymer chain and moving considerably slower than a free bead. The main impact these numbers have on the simulation is the time scale, however it is very important as they are trying to make a case for the timescale and consistency with remodeler activity (Line 233).

Response:

We thank the reviewer for raising this important point. As noted, the choice of diffusivity and viscosity primarily impacts the timescales in our simulations, and we agree that careful justification of these parameters is essential for biological consistency. We reported a diffusivity of $1 \mu\text{m}^2/\text{s}$ ($10^{-8} \text{ cm}^2/\text{s}$) for epigenetic diffusion, which is not the same as the effective diffusion of the chromatin polymer itself. To further elaborate, we define the diffusion of epigenetic marks as the energetically driven redistribution of these marks, without any displacement of the polymer segments themselves. Contrastingly, the diffusion of the chromatin polymer refers to the spatial displacement of individual segments of the chromatin polymer. As discussed in response to the previous question from the reviewer, the effective diffusion constant of the chromatin polymer lies in the range of $D \approx 10^{-3} \mu\text{m}^2/\text{s}$ which matches well with experimentally observed data, as described by the reviewer and previously published work[6].

We still acknowledge that there may be discrepancies reported in values and the ones we find here which do arise from differences in experimental conditions, specific chromatin contexts, or other assumptions. To address this concern and ensure clarity on how choice of the viscosity determines the timescales, we have revised the methods section to include these details and added an expanded discussion in SI1.3.

3. The authors did a commendable job showing how the model interacts with various experimental measurements. However, they overstate their case stating that "reactions and diffusion compete to give rise to chromatin domain" "a result that has not been previously reported". A variety of studies introduce features that reduce the homogeneity of a simple bead spring model. Some of these are densities of GC-content, CtCF sites doi: 10.1038/s41598-017-02923-6; density and distribution of cross-linkers doi: 10.1093/nar/gkx741; different chromatin states open and dumbbell <https://doi.org/10.7554/eLife.60312>). What these authors have implemented is a bead exchange algorithm that is energetically favorable. Based on prior studies, there is every expectation that heterogeneity in the system will arise, and it is no surprise they can tune the energetics to fit experimental data.

Response:

We appreciate the reviewer's thoughtful feedback and recognize that our original wording may have been unclear. We have revised the manuscript to clarify our claims and avoid overstating

our contributions. To the best of our knowledge, this study is the first to address the presence of 3D chromatin “clutches,” as observed through STORM imaging[7]—referred to here as “chromatin domains”—within a molecular dynamics framework that incorporates both diffusion and reaction dynamics.

It is important to distinguish the domains described in our work from those formed by CTCF-mediated chromatin interactions. Unlike CTCF domains, the reaction-driven chromatin domains as predicted by our model persist even after cohesin/RAD21 knockdown[8, 9]. Hence the changes to chromatin organization due to CTCF perturbations will likely have a different effect than what we propose in the current manuscript. Our current work builds on our prior theoretical modeling efforts from our group[10, 11] and expands the focus to highlight the interplay between epigenetic diffusion, reactions, and chromatin organization at a kilobase scale.

We fully acknowledge the significance of prior studies that introduce heterogeneity into bead-spring models, such as those involving GC-content densities, CTCF sites, cross-linker distributions, or different chromatin states. However, our approach emphasizes the distinct role of reaction-driven chromatin domain formation in epigenetic and transcription regulation. For example, while CTCF knockouts are associated with noisier transcriptional states[12], our findings, along with others, show that perturbing histone modifiers (e.g., TSA or GSK treatments) leads to widespread transcriptional changes. This underscores a more direct role for histone modifications in phenotype determination compared to CTCF-mediated compartmentalization and looping. We acknowledge that accounting for CTCF mediated chromatin organizational changes is important, which we will account for more explicitly in our future work.

Reviewer#2:

1. The new model with the reaction-diffusion mechanism nicely explains why compartment domains do not reach full phase separation but maintain as many small domains. A conventional model that does not depend on this reaction-diffusion mechanism would be entropic penalty prevents the polymer from reaching full phase separation with the minimum enthalpy. I guess if the A-A and B-B interaction strengths are tuned lower and lower, at some point, even without the reaction-diffusion mechanism, the chromatin will not reach full phase separation but maintain as many small domains, because this is entropically favored. This conventional model is potentially consistent with the experimental results as well, as the acetylation or methylation reactions could be modeled as changing the A-A and B-B interaction strengths. Could the authors clarify: Is the new model better than the conventional model, or is it an alternative to the conventional model? In either case, the new model should be published as it represents a conceptually different framework that is inspiring to the field.

Response:

We thank the reviewer for their insightful comment and for proposing an energetics-based modeling scheme. We believe our proposed model not only captures the essence of this conventional approach but also builds upon it in significant ways.

- **Out-of-Equilibrium Dynamics:** A major advantage of our model is its inherently out-of-equilibrium nature. It considers the consumption of energy sources, such as ATP to drive the system towards an energetically unfavorable state. This feature is crucial for accurately modeling the

nucleus, an active system driven by energy-consuming processes. The epigenetic reaction mechanism not only breaks detailed balance, which maintains the system in an out-of-equilibrium state but also inhibits the system from coarsening which leads to the formation of finite-sized characteristic chromatin clutch domains (as shown in Fig. 2a). The conventional model suggested by the reviewer is elegant and might be consistent with several experimental results; however, our proposed model extends such a framework by explicitly accounting for the out-of-equilibrium dynamics that are imperative to active systems and essential for domain formation in our case.

- **Consistency with experimentally observed length scales:** It is important to note that the interaction parameters (ϵ_{HH} and ϵ_{EE}) are tuned against experimental data such that they follow the observed spatial distance to genomic distance parameter. Hence, while tuning the A-A and B-B interaction strengths to lower values (as suggested by the reviewer) may indeed maintain chromatin domain formation and prevent them from ripening due to entropic effects, this approach will introduce discrepancies between the observed chromatin organization length scales and the computationally produced length scales. We show the effect of reducing the interaction potentials such that we have lower A-A and B-B interactions in Fig. R2 (corresponding to Fig 1d). This clearly shows that such a model would not fit the observed length scales. In contrast, the potentials in our model are calibrated to replicate experimentally observed scaling relations, ensuring consistency with available data.
- **Incorporation of tuning interaction strengths:** The reviewer's suggestion of modeling acetylation or methylation as modifications to the A-A and B-B interaction strengths is already encapsulated within our framework. For instance, consider a pair of chromatin segments (1 and 2) that currently exhibit an A-A interaction. Through the reaction-diffusion mechanism, these segments can transition to A-B, B-A, or B-B interaction schemes, effectively altering their interaction energy. This mechanism captures the essence of the proposed idea. However, we acknowledge a limitation in our current setup: our model allows for only three discrete attraction energies, whereas multiple epigenetic marks could

Figure R2: Mean spatial distance vs genomic distance for smaller interaction parameters.

imply a broader spectrum of interaction strengths. Expanding this aspect is a promising direction for future.

In summary, while the reviewer's model is highly interesting and conceptually consistent with several experimental findings, we believe our proposed model not only captures its essence but also incorporates additional critical features, such as active dynamics, that make it better suited for capturing chromatin behavior within the nucleus.

2. An ideal experiment that can fully test the validity/contribution of the new model vs. the conventional model to real compartment organization would be: Treat the cells with a cocktail of inhibitors that stops all acetylation and methylation reactions, so that the epigenetic identities along chromatin are "fixed". Then if the new model is correct, the A-B compartments in the cells would reach full phase separation without the reactions. If the conventional model is correct, the A-B compartments in the cells would not reach full phase separation since the entropic penalty is still there. I'm not sure if this experiment is doable but the authors could discuss and suggest this possibility for a future work, as this will generate fundamental insights regarding which model is correct (or how much each model contributes to reality, since they are not completely exclusive to each other).

Response:

We sincerely appreciate the reviewer's thoughtful suggestion. This proposed experiment is indeed a fascinating test for evaluating the validity of our model. As the reviewer notes, however, there are significant practical challenges that make such an experiment unfeasible. Specifically:

1. Administering a potent histone remodeler inhibitor cocktail that halts all acetylation and methylation reactions would cause widespread transcriptional changes. Such extensive disruptions to intracellular transcriptomes and proteomes would make it difficult to attribute any observed effects specifically to the mechanisms being studied. Additionally, any "leakiness" in the inhibitors could introduce unintended effects, complicating efforts to test the theory precisely. Moreover, many inhibitors targeting histone modifications also affect acetylation and methylation of other proteins to some extent, further preventing the isolation of effects to chromatin.
2. As illustrated in Fig. 2a, our model predicts that the time required for full phase separation is on the order of hours for a single chromosome. Scaling this process to the entire nucleus would take even longer. Maintaining cell viability under these conditions would be extremely difficult, if not impossible, further complicating the experiment.

We agree with the reviewer that such an experiment is not feasible with current technology. While this experiment is highly unlikely to be performed, it represents an ideal validation of the proposed model. To address this, we have added the following text to the discussion: *"Decisively validating our model would require inhibiting all epigenetic reactions simultaneously to observe chromatin behavior under such conditions, but this is not feasible in a living cell."*

- In terms of the diffusion, my understanding is the authors allow only diffusion in 1D along the DNA, not in 3D when two regions are in contact. It would be nice if the authors can explicitly make this clear that 3D diffusion is not included in this model.

Response:

We thank the reviewer for bringing this to our attention. To clarify, our model exclusively considers 3D epigenetic diffusion and does not include 1D epigenetic diffusion along the DNA. We have revised the methods section to explicitly highlight this choice and ensure that it is clear to the reader by adding the following text: “We note that this diffusion is performed in 3D and not along the polymer strand, as shown explicitly in Fig S4.”

- The authors showed in simulation that upon changes of reaction rate, compartment identity changes are enriched at domain boundaries. This phenomenon needs a better explanation. Based on the model description, random beads are chosen to be converted to heterochromatin or euchromatin beads with defined rates. How come the chromatin regions at domain boundaries are more often altered? My guess is: after a random bead is converted, the new bead identity (in a domain of opposite identity) can diffuse, and when it diffuses to the boundary, it is “fixed” there due to the favorable energy. As a result, after simulation of a time period, more alterations are “fixed” at the boundaries. Is this guess correct?

Response:

We thank the reviewer for highlighting this important aspect of our model, which could benefit from further clarification. The reviewer’s interpretation is correct: beads can randomly change their epigenetic identity due to epigenetic reactions. Once their identity changes, they are likely to diffuse out of an unfavorable environment and become stabilized at the domain

Figure R3: Dynamics favor boundary enrichment of epigenetic changes. Following a methylation event, the altered bead diffuses through the chromatin domain. Its most stable position is at the boundary of the existing domain, as the energy landscape is more favorable due to heterochromatin’s affinity for similar chromatin types.

boundary, where the energy landscape is more favorable. We show this explicitly through schematic Fig. R3. In the given example, we show the diffusion of a heterochromatin bead after its *de novo* establishment through a methylation step. As is clearly seen, through the diffusive process, the newly established epigenetic mark will diffuse out of a euchromatin rich environment and get stabilized at the boundary of a domain boundary.

To provide a clearer explanation of this phenomenon, we have included this discussion at the end of Section 3b along with the schematic (Fig. R3 reproduced as Fig. S24) to illustrate the detailed process.

5. Line 697: After “Heo et al” a citation number is missing.

Response:

Thank you for pointing out the missing citation. We have added it.

6. Line 764-765: “When undergoing extracellular: ...” A word is missing here, likely “stimulation”.

Response:

We have added the missing word, and stimulation fits the bill perfectly.

7. Line 783-785: “Therefore, smaller domains can act as domain boundaries themselves and give rise to lasting memory characteristics”. I thought the results above just showed that smaller domains do NOT give rise to lasting memory and can disappear upon perturbation, and upon removal of perturbation do not reappear at the same region but at nearby region. The bigger domains give rise to lasting memory and at least partially recover.

Response:

We thank the reviewer for pointing out this confusion and appreciate the opportunity to clarify. First, we acknowledge that the term "memory" can be loosely interpreted, so we have made efforts to define it more precisely. In Fig. 7c, we explicitly categorize chromatin segments based on their response to perturbations:

- a. “Conserved” segments: Those that retain their original epigenetic state throughout the simulation.
- b. “Memory” segments: Those that change their epigenetic state in response to external stimuli and do not revert to their original state upon returning to control conditions.
- c. “Recovered” segments: Those that change their epigenetic state under perturbation but revert to their original state once the perturbation is removed.

This classification allows us to quantify the degree of memory formed in the system, akin to the concept of hysteresis in physics.

In the specific case referred to by the reviewer, smaller domains indeed exhibit "memory" because, following perturbation, they do not recover their original state upon the removal of stimuli. This behavior reflects a memory of the perturbation, distinguishing it from the elasticity observed in larger domains, which are more likely to partially recover their original state. Had the smaller domains fully recovered, they would be considered elastic and would show no memory of the intermediate perturbation. We hope this explanation resolves the confusion and provides a clearer understanding of our results.

Reviewer #3:

Major Comments:

- 1) The concept of diffusion of epigenetic marks: At the molecular level, epigenetic marks being covalent modifications on histone proteins or DNA, established by enzymatic reactions (active). While as purely mathematical model its interesting, but since being contextualized to chromatin,

I find it hard to understand the basis (biological context or origin) for having “diffusion of epigenetic marks” or the use of Kawasaki dynamics here. For instance, several experiments have shown that heterochromatin domains to decay completely when enzymes are mutated (Ex: Audergon et al. Science 2015, Ragnathan et al. Science 2015). The redistribution/diffusion of epigenetic marks is a puzzling choice.

Response:

We thank the reviewer for raising this important point of the biological origin for the diffusion of epigenetic marks and the use of Kawasaki dynamics in our model. As the reviewer insightfully notes, epigenetic marks are covalent modifications of histone proteins, established through enzymatic reactions, which are active processes. We agree with this characterization but would further emphasize that these processes are also neighborhood-dependent, where local chromatin epigenetic states influence enzymatic activity and subsequent mark deposition. An intuitive explanation of how the activity of epigenetic enzymes is neighborhood dependent is as follows:

We begin with the fundamental principle that interactions between similar chromatin regions (e.g., heterochromatin-heterochromatin) are more energetically favorable than those between dissimilar regions (e.g., heterochromatin-euchromatin). Consequently, breaking interactions between similar states requires more energy compared to breaking interactions between dissimilar states, hence, making it easier for epigenetic enzymes to modify marks that are dissimilar to their neighbors. This neighborhood-dependent activity creates a driving force for epigenetically similar marks to cluster together, promoting phase separation of chromatin regions. However, these enzymes, specifically histone epigenetic remodelers, are active, which implies that they can drive energetically unfavorable changes in epigenetic states through energy consumption, for example, through dephosphorylation[13]. Therefore, the balance between the enzymatic activity and the neighborhood-dependent energy landscape ultimately determines the extent of epigenetic clustering. This interplay governs the emergent chromatin organization observed in our model.

We incorporate the *diffusion of epigenetic marks* into our model to capture this neighborhood-dependent activity in an effective manner. Below, we provide a detailed justification for our mechanistic choices, specifically focusing on the rationale for introducing epigenetic diffusion using a kinetic Monte Carlo (KMC) framework, which enables us to explore the emergent dynamics of chromatin organization in a biologically informed context:

Mechanism of Histone Epigenetic Remodeler Activity and a corresponding KMC model

The activity of the histone epigenetic remodeler depends on the following factors:

1. Epigenetic Makeup of the Neighborhood: At a given genomic locus, the effect of activity of histone epigenetic remodelers (e.g., HDAC, HMT, HAT, or HDM) is dependent on epigenetic marks of the neighborhood and the energy required by a remodeler to switch a mark (e.g., from methylation to acetylation or vice versa) depends on the states of its surroundings. For example, the energy required to convert a methylated (red) locus to

Figure R3: A neighborhood-dependent epigenetic reactions formulation gives rise to diffusion and reactions. a) Epigenetic transition rates which depend on the epigenetic makeup of the neighborhood and the activity of enzymes. b) Neighborhood dependent reactions lead to coarsening. c) Finite sized domains are formed at steady state. d) Intermediate steps show a resultant conservative exchange of epigenetic marks. e) Effective exchange of epigenetic marks, akin to diffusion of epigenetic marks shown through a two step process.

More rigorously, within a KMC framework, this dependency is reflected in the probability of epigenetic state transitions. The probability of a chromatin bead changing from red to blue ($P_{R \rightarrow B}$) can be defined as:

$$P_{R \rightarrow B} \propto e^{-\beta \Delta E_{R \rightarrow B}}$$

And similarly for blue to red ($P_{B \rightarrow R}$):

$$P_{B \rightarrow R} \propto e^{-\beta \Delta E_{B \rightarrow R}}$$

Here, β corresponding to the effective inverse temperature (which may be higher than the ambient temperature due to activity[14]), represents the stochasticity of the interconversion process. We note that this energetically driven epigenetic remodeling mechanism will give rise to epigenetic coarsening as it would tend to bring likes together (Fig R3b).

2. Role of the intrinsic activity of the histone epigenetic remodeler enzyme: The rate of epigenetic changes also depends on the nuclear concentration and activity of histone remodelers. For example:
 - o Higher concentrations of histone methyltransferases (HMTs) in the nucleus will result in an increased global methylation rate.
 - o Even at identical concentrations, different remodelers can exhibit varying levels of activity due to differences in enzymatic efficiency or and size[9].

acetylated (blue) (denoted as $\Delta E_{R \rightarrow B}(x)$, where x denotes the position of the mark) will be lower in a blue-rich environment compared to a red-rich one. This is because, in a red rich environment, the enzyme will have to break two red-red interaction bonds (corresponding to HP1 interaction and other physical interactions) but these interactions do not impede the epigenetic interconversion in a blue rich case (Fig. R3b). Similarly, the reverse process ($\Delta E_{B \rightarrow R}$) is also neighborhood dependent. This introduces a local energy landscape that dictates remodeler activity.

To account for such variations, we introduce a multiplicative term α to modify the blue-to-red transition probability:

$$P_{B \rightarrow R} \propto \alpha e^{-\beta \Delta E_{B \rightarrow R}}$$

The value of α reflects the nuclear remodeler concentrations and their activity levels, which account for microenvironmental and drug driven changes, and allows our model to incorporate treatment-specific variations.

With this modeling scheme (Fig. R3a), we conducted KMC simulations and observed the following key results:

1. Steady-State Domain Formation:
 - At steady state, the KMC simulations generate finite-sized domains, similar to those observed in our experimental and computational results (Fig. R3c).
2. Intermediate Dynamics:
 - During intermediate stages (e.g., Fig. R3d,e), the simulations display effective, conservative exchange of epigenetic marks, resembling Kawasaki dynamics.

Based on these observations, our primary insight is that the action of histone epigenetic modifiers can be decomposed into two distinct components:

- A conservative component corresponding to neighborhood-dependent diffusion-like behavior.
- A non-conservative component representing uniform, reaction-like processes.

A corresponding analytical derivation of this approach is provided in SI1.6-1.7 of Kant et. al[11]. As we have shown in the publication, the neighborhood dependence of the epigenetic reactions give rise to an effective epigenetic diffusion constant of the following form:

$$D_{epigenetic\ marks} \approx (\delta x)^2 \Delta \Gamma$$

where δx corresponds to the nucleosome spacing and $\Delta \Gamma (\propto \Gamma_{ac} e^{(\epsilon_{HH} - \epsilon_{EH})/kT})$ depends on the homotypic and heterotypic interactions between the epigenetic states.

This decomposition forms the foundation for introducing a **neighborhood-dependent conservative “diffusion of epigenetic marks” mechanism** and a **uniform non-conservative “epigenetic reaction” mechanism** in our model. The use of **Kawasaki dynamics** is specifically motivated by the need to capture conservative, energetically driven mechanisms. By breaking down the system into these components, we achieve two key advantages:

1. A clear distinction between energetically driven (conservative) and active (non-conservative) processes.
2. The ability to model treatment-induced changes uniformly across the system without introducing positional bias.

We hope this clarifies the reviewer’s concerns about the motivation of the proposed dynamics. We have included this discussion in SI1.7.

2) By construction the model is a competition between diffusion and active reactions, this makes all perturbation on rates affect the domain boundaries as nicely illustrated in Figure 2. Alternatively, previous models based on spreading of epigenetic marks from a nucleation site present a mechanism where domain boundaries are affected on changing rates of ‘epigenetic remodelers’ as well (Ancona et al. PRE 2022, citation [45,54], for instance, figure 2 in citation

[45]). Discussing and contrasting with alternate possibilities/mechanisms previously reported is essential.

Response:

We thank the reviewer for this comment. Firstly, we note that specifically epigenetic spreading occurs over multiple generations and can involve many players[15-17], unlike what we propose here, where the central players are histone epigenetic remodelers. But we acknowledge that such phenomenon can occur at smaller timescales through mechanisms yet unknown and hence we have added the following comparisons with existing literature in the discussion section:

“We acknowledge that prior studies [18-20] have described spreading-driven boundary enrichment mechanisms, where spreading originates from a specific genomic locus and is shaped by the prescribed dynamics of epigenetic modifiers and other major players. Even though such mechanisms predict changes at the domain boundaries, our work departs from these earlier approaches in two key aspects:

- 1. Unlike previous models, which primarily focus on the spreading of epigenetic marks, our study emphasizes the spatial relationship between boundary domains and chromatin clutch domains as observed through STORM. Our findings demonstrate a direct relation between domain boundaries and the physical organization of chromatin clutch domains, a phenomenon we observe experimentally and validate through modeling. This aspect introduces a novel layer of biological relevance, connecting domain dynamics with the underlying chromatin architecture, which has not been explored in prior models.*
- 2. Our model employs a deconstructed physical framework, where the out-of-equilibrium kinetics is governed solely by reaction rates, which break detailed balance. In contrast, previous studies rely on both active writer and active eraser mechanisms operating in tandem, making their models inherently more complex and departing it from the proposed mechanism in the involved timescales as well. This distinction allows our model to be more straightforward in its physical interpretation while still offering robust biological insights.*

Our approach presents a mechanistic perspective by isolating the contributions of reaction-driven kinetics, providing clarity on how these mechanisms influence heterochromatic domain formation. By offering a physically and biologically interpretable framework, we believe our study complements existing literature and offers a novel mechanistic understanding of boundary positioning and dynamics.”

Detailed Comments

Figure1:

1) For better reproducibility, include the parameters for figures d and e in the figure description ($\epsilon_{EE} = 0.3$ and $\epsilon_{HH} = 1$). Why was a ‘hit-and-try’ method used instead of exploring the

parameter space (at least around the hit) showing the fit-score. This would also inform on how sensitive the system is to the parameter.

Response:

- We have added the parameters to the Fig. 1d, e and in the description.
- In our study we have used an iterative method since the objective of the study was not to optimize accuracy but maintain the length scales (as observed through FISH experiments, Fig 1d) and hierarchical structures (as observed through Hi-C maps, Fig 1e).
- We have included the fits for a few potential pairs in Fig. R4 (also added to the Fig. S26). Considering our constraint of $\epsilon_{EE} \leq \epsilon_{HH}$ we find that that the fitting is not very sensitive to the potentials used around the chosen pair. We specifically choose 0.3-1 as higher potentials tend to lead to a globule-like state of the chromatin polymer (as observed in [21]), which is not observed through imaging.

Spatial distance vs. Genomic distance of Chr20 with different potentials

Figure R4: Fitting experimental data with simulations for different attractive potential pairs.

2) Figure S2, Random-walk initialization potentially gives rise to knots in the system. Is there a change in the kinetics if an alternate knot-free initialization is used?

Response:

As the reviewer points out, knots in polymer chains can significantly influence their relaxation dynamics (Figure S2). For instance, previous works have shown that knotted polymers exhibit different stretching responses compared to unknotted ones, with the ratio of their extensions varying nonmonotonically with applied force[22]. Additionally, the presence of knots can lead to mechanical scission under tension, indicating that knotted polymers may have different mechanical relaxation behaviors compared to unknotted ones[23]. In our study, we employed multiple initial conformations for the polymer chains, which potentially include both knotted and unknotted

Figure R5: Relaxation of the polymer starting from a linear polymer configuration.

we employed multiple initial conformations for the polymer chains, which potentially include both knotted and unknotted

structures, with knotting at distinct genomic loci. We do this to ensure a comprehensive sampling of the polymer conformational space. Our approach inherently introduces heterogeneity in the system, as each conformation may exhibit distinct relaxation kinetics due to the presence or absence of topological constraints, including knots. By averaging the results across simulations with diverse initial conformations, the unique relaxation behaviors of knotted configurations may dilute, leading to ensemble properties that resemble those of unknotted systems. This averaging effect can mask the distinct kinetics associated with knotted polymers, resulting in relaxation plots that do not prominently display the slower relaxation dynamics induced by topological constraints.

To address the reviewer's concern, we conducted additional relaxation using knot-free initial conformations, wherein we started with a linear polymer and equilibrated it. We note here that even though no knots are present in the initial configuration, knots might be introduced due to the relaxation process as reported previously[24]. The relaxation plots from these simulations (Fig. R5) were compared to those obtained from the original set. Our analysis reveals that the ensemble-averaged relaxation kinetics remained consistent between the two sets of simulations. This consistency suggests that the heterogeneity introduced by the random-walk initialization, which potentially includes knotted conformations, does not significantly alter the overall relaxation behavior of the system.

3) Timescales: Histone turnover rate (\sim per hour) in Alabert, C et al. Genes and development 2015, hours to days in citation [69], Aimee M Deaton et al. elife 2016 (faster turnover rate) and HMT based reaction rates (min^{-1}) in Newar et al PLoS CB 2022 seems to show a lot of variability depending on cell-type/stage. Since there is no explicit 3D organization dependent spreading, how would the ratio of epigenetic reaction rate and polymer dynamics (currently one reaction per 10^3 simulation step) affect the system? Maybe in the supplementary, include a figure to illustrate the effect.

Response:

We thank the reviewer for highlighting the important aspect of the timescales of reaction rates relative to chromatin diffusion rates. This is a crucial consideration, and we have it in our analysis in Section SI2.2. Specifically, we provided simulation runs for systems with both high (one reaction step per simulation step) and very low reaction rates (One reaction step per 10^5 simulation steps), while keeping other parameters constant. As illustrated in the figure S12, very high reaction rates lead to a randomization of epigenetic marks, whereas very slow reaction rates result in a ripening-like state.

We appreciate the references suggested by the reviewer and have now included them in the manuscript Section 3a to further contextualize and enhance the relevance of this discussion.

Figure 2:

4) Mention the reaction rates used in the figure description. Why does it require the use of DBSCAN to compute R_g , isn't the spatial x,y,z coordinates of the monomers known?

Response:

- Reaction Rates in Figures: We have included the reaction rates used in the figure description to address the reviewer's suggestion.

- Use of DBSCAN for Radius of Gyration (R_g) Calculations: For our analysis, we are not calculating the radius of gyration (R_g) of the entire chromosome polymer, which would only require the spatial coordinates of all monomers. Instead, we specifically calculate the R_g of spatially distinct 3D heterochromatin-rich domains. Identifying these domains in 3D space requires a robust clustering approach, for which we used the DBSCAN algorithm. This method provides an effective basis for identifying distinct spatial domains within the chromatin structure. We then calculate the R_g for each identified domain separately.

5) Since the time mapping has been already done, it would be more informative if the x-axis of figure a is in real time, the conversion maybe be mentioned in the description.

Response:

This is a great suggestion from the reviewer. We have changed the labeling of our axes to reflect time in real units.

6) Irrespective of the initial distribution, based on the ratio of reaction rates (me/ac), can we predict the number of distinct, persistent separated domains or is it stochastic? While the model can sustain distinct, separated domains, won't the domains be of the same size at steady state?

Response:

We thank the reviewer for highlighting this important aspect. The number of domains and their size distributions are inherently stochastic, governed by a combination of factors. Notably, these distributions are strongly influenced by the initial conditions, such as the input ChIP-seq or Hi-C A/B compartmentalization, as well as the intrinsic noise arising from the stochasticity of the epigenetic reactions. These parameters dictate the dynamic processes underlying domain formation and persistence.

At steady state, the domains do not settle into uniform sizes but instead exhibit a size distribution. In our simulations, this distribution aligns with a heavy tailed distribution (like a lognormal profile), as demonstrated by the multiple size distribution curves obtained (Fig. 3c, 4c and 6e).

Figure 3:

7) Quantify the fraction of monomers that retain the same state between t=0 and t=2 hours, mention it in the description for figure a and d.

Response:

We have added this information to the caption of Fig 3.

8) Show the pairwise distance map comparison for the initial and final state for the case shown in figure 3a, it would help the comparison. Also, in the supplementary plot a difference map to quantify and highlight the difference between the timepoints. Mention the energy parameters in the main text.

Response:

We thank the reviewer for the suggestion.

- The point-to-point distance maps for the initial and final states corresponding to Fig. 3a are in Fig. S10, which we believe addresses the reviewer's suggestion.
- To further clarify, we have now included a difference map as Fig. S27 (Fig. R6), which highlights the regions where the most significant changes occur over time.
- We have incorporated the energy parameters into the main text for better context and completeness.

Figure R6: Difference plots for chromosome 19 after 2 hours. The left figure shows control setup, and the right figure shows the acetylated setup. We have subtracted the distance matrix of the final from the initial. The green boxes show the position of the initial domains.

9) Extending question 6, doesn't figure c show a peaked distribution upon the ratio of rates being slightly skewed, implying the model predicts most of the domains to have uniform size? Does the mean domain size remain the same after the perturbation or what is the difference?

Response:

We thank the reviewer for their observation. While Fig. 3c shows a peaked distribution in both the control and acetylated cases, this does not imply that the domains are uniform in size. The peak represents the most probable domain size, but the overall distribution remains heavy-tailed. It is worth noting that even in the control case, which reflects the input ChIP-seq/Hi-C epigenomic annotation, the distribution exhibits a strikingly similar, heavy-tailed shape. This indicates that while many domains cluster around the peak, there is substantial variability, particularly in the tail of the distribution, which captures the presence of larger domains (e.g., centromeric regions and other large heterochromatin rich structures).

Following a perturbation, the mean domain size does not remain constant but shifts depending on the extent and type of the perturbation. For example, in Fig. 3c, increased acetylation leads to an ~11% decrease in the mean domain size (as described in Section 3c). Given the skewed nature of the distribution, even small shifts in the tail can have a significant impact on the mean. These effects underscore that domain sizes remain heterogenous and do not converge to uniformity, even after perturbations.

Figure 4:

10) The section heading "STORM imaging confirmed chromatin domain size scaling in agreement with model prediction" is misleading because it doesn't confirm the scaling argument presented in equation 1.

Response:

We have changed our title to: "*STORM imaging shows qualitative agreement with model predicted chromatin domain scaling,*" which better captures the result shown.

11) TSA treatment shows a decrease in heterochromatin, upon increasing acetylation impairing heterochromatin is expected (Zofall M. et al. 2022 NSMB, Sahu R K et al. Mol Cell 2024, among many other studies) and is that a key prediction?

Response:

We thank the reviewer for their question. The decrease in the average heterochromatin content itself is not the central prediction of the model even though the fact that our model is able to capture it lends validity to our predictions. Instead, the key prediction lies in the changes to the domain sizes of heterochromatin-rich histone clutches that result from this decrease in heterochromatin content. Both the model and experimental data consistently demonstrate that the mean domain size decreases following TSA treatment. However, it is worth noting that other outcomes could theoretically occur. For instance, if heterochromatin loss were confined to smaller domains, the mean domain size could increase instead.

Our model further highlights that heterochromatin loss does not occur uniformly over domains but occurs predominantly at domain boundaries. This prediction is tested and supported by the patterns observed in the Hi-C maps, which reveal structural changes in domain organization following perturbation.

12) In the simulation (figure c), what is the increase in acetylation rate and how was this value chosen? Isn't there an acetylation rate change that would give closer to 8% change in mean? Wouldn't the behavior be the same if methylation rate is reduced? So, can an alternate perturbation inhibiting methyltransferase be used to validate the model?

Response:

- In the current manuscript, we determined the final reaction rate using an iterative strategy, testing various possible changes in reaction rates until we identified a rate that yielded a similar decrease in mean domain size between the observed and simulated results. For the case of Fig. 4c, we selected an acetylation rate (probability) of 0.6.
- While it is possible to identify an acetylation rate that results in a closer match to the observed 8% change in mean, the primary objective of this manuscript is to elucidate the underlying biophysical regulatory mechanisms rather than optimize for numerical accuracy. Achieving a perfect match would require significantly more computational effort due to the stochastic nature of the system and the time-intensive simulations. Therefore, we opted for the current approach to balance these considerations.
- As discussed in detail in the Methods section, the model treats acetylation and methylation as complementary processes. Consequently, the observed results could also be interpreted as a decrease in methylation. But we note that the activity of different histone epigenetic remodelers might differ and hence inhibiting complimentary ones won't have an equal and opposite effect.

- We appreciate the reviewer’s suggestion regarding the effect of a histone methyltransferase (HMT) inhibitor. Indeed, this approach aligns with what we have demonstrated in Section 3e, where we captured the experimental observations of hMSCs following GSK treatment, which functions as an HMT inhibitor.

13) Is the take-away from the section “averaging over cell population leads to fewer epigenetic compartmental changes” that averaging or bulk data misses out ON cell-to-cell variability? Isn’t that the basis of averaging, to see changes that are present more frequently in population?

Response:

We thank the reviewer for raising this insightful question. There are two key takeaways from this section:

1. **Epigenetic changes primarily occur at domain boundaries.**
2. **Additional changes occur elsewhere but are not captured in sequencing data due to cellular heterogeneity.**

While we agree with the reviewer that averaging over cell populations inherently masks cell-to-cell variability, we believe it is important to highlight the consequences of this heterogeneity masking. Specifically, cellular heterogeneity can create significant discrepancies between imaging and sequencing results. For example, in our TSA treatment experiments, chromatin reorganization is clearly visible under STORM imaging, yet these changes are not evident in Hi-C maps. This discrepancy occurs because Hi-C maps average out cell-to-cell variability, effectively erasing localized changes that are present only in subsets of cells.

This limitation of sequencing-based approaches has broader implications for data interpretation. For instance, relying solely on Hi-C and RNA-seq data post-TSA treatment might suggest that chromatin organization, which appears minimally perturbed in Hi-C observations, plays little to no role in transcriptional changes (despite significant shifts observed in RNA-seq data). However, our work demonstrates that deeper analyses of Hi-C data—prompted by structural changes observed via STORM imaging—reveal that chromatin organization plays a pivotal role in shaping cell phenotypes.

We hope this response addresses the reviewer’s concerns and underscores the importance of our integrative approach in bridging the gap between imaging and sequencing data while emphasizing the limitations of population-averaged measurements. Additionally, we would like to point out that our manuscript explicitly highlights these two main results through the following points made towards the end of the section:

“In summary, our model highlights two key results: 1) While we observe noticeable changes in STORM imaging, the Hi-C data reflect only those changes that are present for the majority of the cells while not capturing changes that are shown by smaller subpopulations. 2) The compartment boundaries, as confirmed by Hi-C observations, emerge as regions of heightened sensitivity to epigenetic reaction alterations, where transitions in chromatin compartmentalization are favored owing to the enacting kinetics.”

14) Can the shift be attributed to large domains and small domains upon TSA treatment, while intermediate domains remain unaffected as the analysis in figure 3 would suggest? In figure c the shape of the distribution experimental TSA and simulated TSA seems quite different, why is that

so?

Response:

We thank the reviewer for their insightful question.

- While it might appear that the shift is confined to large and small domains, we would like to direct the reviewer's attention to Fig. S16, where we present a detailed distribution of the domain sizes. This figure clearly shows that domains across all size ranges undergo changes upon TSA treatment. However, we agree that the changes at the upper extremes are more pronounced compared to those near the mean.
- Regarding the differences between the experimental and simulated TSA distributions in Fig. 3c, we note that the simulations were carried out on chromosomes 18–21 due to computational constraints. We chose these chromosomes as they give us a wide variety wherein chromosome 19 is a gene-rich and chromosome 18 is gene poor chromosome while 20 and 21 represent typical gene densities. These smaller chromosomes lack the larger heterochromatin-rich contiguous regions typically found near the centromeres and telomeres of larger chromosomes. Additionally, heterochromatin and euchromatin regions from different chromosomes can come together to give rise to larger domains which our model cannot account for. As a result, while our simulations capture the general trends in the experimental data, they miss the maxima of the observed chromatin domain sizes.

Figure 5:

15) 30% of DE genes are within 300 kb of a boundary element, in the model/simulation does TSA treatment change that happens at the boundary (300kb) as opposed to more central?

Response:

In our model, as shown in Fig. 4h, the observed epigenetic changes and the resulting chromatin-chromatin interaction changes, corresponding to TSA treatment, occur predominantly near boundary regions rather than in the central regions of chromatin domains. The experimental Hi-C data reveals that these changes are primarily localized to boundaries, typically within a few hundred kilobases, while the regions at the center of domains exhibit much fewer alterations. This is not the case with the experimentally observed RNA-seq data where transcriptional changes which are prominent in the regions away from the boundaries as well (as seen in Fig. S21).

We attribute this to the fact that, while chromatin organization and epigenetic changes can prime the system for transcription, the actual transcriptional response also depends on additional factors such as the availability of specific transcription factors and RNA polymerases. These factors are not explicitly included in our model, as it currently focuses on the accessibility and epigenetic components.

Figure 6:

16) The claim that polymer model captures chromatin re-organization in response to changes in microenvironmental stiffness is misleading because the polymer model only acts through methylation/acetylation rate irrespective of the mechanism by which this change is brought about by (chemically or mechanically induced). Mechanistically the model doesn't link environmental

stiffness change to organization rather plays only on acetylation/methylation rate as input.

Response:

We thank the reviewer for their observation and agree that the current model does not directly predict chromatin organization from cellular stiffness or contractility. Instead, it uses the observed effects of changes in contractility on epigenetic modifier activity to infer trends in reaction rates, which are then used to predict chromatin organization trends. **To ensure we do not overstate our claims, we have revised the introducing paragraph to include the following text which clarifies this limitation: “The current model predicts chromatin reorganization based on observed changes in epigenetic reaction dynamics in response to variations in substrate stiffness.”**

We would also like to point out that our current model can be coupled with a previous publication from our lab[25], which establishes an empirical relationship between nuclear histone remodeler activity and cellular contractility. This coupling would allow direct predictions of chromatin organization from cellular stiffness. While this approach lies beyond the scope of the current manuscript—due to the complexities of measuring cellular contractility—we highlight this potential extension as a promising future direction. With appropriate modifications, our model could be applied in a broader context, enabling direct computation of chromatin reorganization from substrate stiffness. We expand on this idea in the concluding paragraph of Section 3e.

Figure 7:

17) Epigenetic memory is generally used to refer to retaining epigenomic state post-replication (in the context of heritability). It would be interesting to see how incorporating replication (dilution of methylation/acetylation) would affect the behavior.

Response:

We thank the reviewer for this insightful comment. We have also considered the potential impact of incorporating replication dynamics, including the dilution of methylation and acetylation marks, into our model. While this lies beyond the scope of the current manuscript, we agree that it is an important avenue for future exploration. **To address this, we have added a discussion along the following lines in the discussion section:**

Temporally, we stick to the changes within a single cell cycle. Integrating replication dynamics[26] and epigenetic spreading mechanisms[27] will reveal long-term chromatin architecture changes, which are crucial for understanding cancer metastasis and development. This will require us to incorporate the role of central proteins such as condensin motors which drive sister chromatid formation and eventual cellular division[28]. HP1 proteins have also been shown to be essential in determining the epigenetic stability of histone modifications across cell cycles[29].

We believe that introducing replication dynamics in the current study could deflect from its primary focus, but we are actively working on extending our model to include these processes in future work.

18) A remark on citation: Earlier works Sandholtz S H.et al. PNAS 2020 and citation [45] illustrate chromatin organization and limited enzymes allow stable epigenetic memory.

Response:

We thank the reviewer for pointing out these citations. We have added it at appropriate places now. It is worth noting that the suggested citation pertains to budding yeast. While the general principles may apply across organisms, the A/B compartmentalization patterns in yeast differ from those in humans, and thus not all principles may be directly transferable.

Discussion section:

19) The discussion section would benefit from a detailed section on the limitations of the model, possible perturbations the model does not capture.

Response:

We have expanded on the limitations of the model and the various effects and perturbations it is unable to capture towards the end of our discussion section. We now have an explicit limitations and future directions sections which is reproduced below for convenience:

“Limitations of the model and future directions: While our model has proposed key biophysical mechanisms governing chromatin organization and gene regulation, future refinements can address current limitations. In the current setup, we have considered a copolymer simplification which is a major hinderance in the capabilities of the model. Hence, incorporating multistate chromatin models that account for different epigenetic flavors (such as ChromHMM[30] or MEGABASE[31]), chromatin–lamina interactions[32], and sequence-dependent epigenetic marking[33] will increase the predictive power. Additionally, the influence of other molecular machinery on chromatin organization has not been considered. For instance, RNA Polymerase II-driven supercoiling[34] is known to play a crucial role in chromatin organization. Similarly, condensing proteins such as HP1[35], transcription factor binding[36], and cohesin-driven loop extrusion[37] significantly impact the spatial positioning of genomic loci. These factors are essential in shaping the 3D chromatin organization, which is central to the epigenetic and transcriptional shifts predicted by the model.

Lastly, the model also has spatial as well as temporal limitations. All the simulations we have performed in this study are over a single chromosome, while it is well known that regions in between chromosomes can play the role of transcription hubs and hence are central to determining expression. Temporally, we capture and predict changes within a single cell cycle. Integrating replication dynamics[26] and epigenetic spreading mechanisms[27] will reveal long-term chromatin architecture changes, which are crucial for understanding cancer metastasis and development. This will require us to incorporate the role of central proteins such as condensin motors which drive sister chromatid formation and eventual cellular division[28]. HP1 proteins have also been shown to be essential in determining the epigenetic stability of histone modifications across cell cycles[29]. Lastly, exploring the interplay between chromatin organization and nuclear signaling pathways will provide a more comprehensive understanding of cell fate determination. By building upon our foundation, future studies can investigate the role of chromatin organization in disease mechanisms, ultimately informing novel therapeutic strategies. The integration of experimental and theoretical approaches will be essential for achieving a holistic understanding of chromatin organization and its role in governing cell fate.”

References:

1. Nicklas, R.B., *Measurements of the force produced by the mitotic spindle in anaphase*. J Cell Biol, 1983. **97**(2): p. 542-8.
2. Fisher, J.K., et al., *DNA relaxation dynamics as a probe for the intracellular environment*. Proc Natl Acad Sci U S A, 2009. **106**(23): p. 9250-5.
3. Cai, S., et al., *Poroelasticity of a covalently crosslinked alginate hydrogel under compression*. Journal of Applied Physics, 2010. **108**(11).
4. Brahmachari, S., et al., *Temporally Correlated Active Forces Drive Segregation and Enhanced Dynamics in Chromosome Polymers*. PRX Life, 2024. **2**(3): p. 033003.
5. Caragine, C.M., S.C. Haley, and A. Zidovska, *Surface Fluctuations and Coalescence of Nucleolar Droplets in the Human Cell Nucleus*. Phys Rev Lett, 2018. **121**(14): p. 148101.
6. Salari, H., M. Di Stefano, and D. Jost, *Spatial organization of chromosomes leads to heterogeneous chromatin motion and drives the liquid- or gel-like dynamical behavior of chromatin*. Genome Res, 2022. **32**(1): p. 28-43.
7. Ricci, M.A., M.P. Cosma, and M. Lakadamyali, *Super resolution imaging of chromatin in pluripotency, differentiation, and reprogramming*. Curr Opin Genet Dev, 2017. **46**: p. 186-193.
8. Li, W.S., et al., *Chromatin packing domains persist after RAD21 depletion in 3D*. bioRxiv, 2024: p. 2024.03.02.582972.
9. Miron, E., et al., *Chromatin arranges in chains of mesoscale domains with nanoscale functional topography independent of cohesin*. Sci Adv, 2020. **6**(39).
10. Heo, S.J., et al., *Aberrant chromatin reorganization in cells from diseased fibrous connective tissue in response to altered chemomechanical cues*. Nat Biomed Eng, 2023. **7**(2): p. 177-191.
11. Kant, A., et al., *Active transcription and epigenetic reactions synergistically regulate meso-scale genomic organization*. Nat Commun, 2024. **15**(1): p. 4338.
12. Hafner, A., et al., *Loop stacking organizes genome folding from TADs to chromosomes*. Mol Cell, 2023. **83**(9): p. 1377-1392 e6.
13. Bahl, S. and E. Seto, *Regulation of histone deacetylase activities and functions by phosphorylation and its physiological relevance*. Cell Mol Life Sci, 2021. **78**(2): p. 427-445.
14. Ganai, N., S. Sengupta, and G.I. Menon, *Chromosome positioning from activity-based segregation*. Nucleic Acids Res, 2014. **42**(7): p. 4145-59.
15. Cutter DiPiazza, A.R., et al., *Spreading and epigenetic inheritance of heterochromatin require a critical density of histone H3 lysine 9 tri-methylation*. Proc Natl Acad Sci U S A, 2021. **118**(22).
16. Obersriebnig, M.J., et al., *Nucleation and spreading of a heterochromatic domain in fission yeast*. Nature Communications, 2016. **7**(1): p. 11518.
17. Hall, I.M., et al., *Establishment and maintenance of a heterochromatin domain*. Science, 2002. **297**(5590): p. 2232-7.
18. Abdulla, A.Z., C. Vaillant, and D. Jost, *Painters in chromatin: a unified quantitative framework to systematically characterize epigenome regulation and memory*. Nucleic Acids Res, 2022. **50**(16): p. 9083-9104.

19. Katava, M., G. Shi, and D. Thirumalai, *Chromatin dynamics controls epigenetic domain formation*. Biophys J, 2022. **121**(15): p. 2895-2905.
20. Ancona, M., D. Michieletto, and D. Marenduzzo, *Competition between local erasure and long-range spreading of a single biochemical mark leads to epigenetic bistability*. Physical Review E, 2020. **101**(4): p. 042408.
21. Shi, G., et al., *Interphase human chromosome exhibits out of equilibrium glassy dynamics*. Nat Commun, 2018. **9**(1): p. 3161.
22. Caraglio, M., C. Micheletti, and E. Orlandini, *Stretching Response of Knotted and Unknotted Polymer Chains*. Phys Rev Lett, 2015. **115**(18): p. 188301.
23. Zhang, M., et al., *Mechanical scission of a knotted polymer*. Nat Chem, 2024. **16**(8): p. 1366-1372.
24. Tubiana, L., et al., *Spontaneous Knotting and Unknotting of Flexible Linear Polymers: Equilibrium and Kinetic Aspects*. Macromolecules, 2013. **46**(9): p. 3669-3678.
25. Alisafaei, F., et al., *Regulation of nuclear architecture, mechanics, and nucleocytoplasmic shuttling of epigenetic factors by cell geometric constraints*. Proc Natl Acad Sci U S A, 2019. **116**(27): p. 13200-13209.
26. Stewart-Morgan, K.R., N. Petryk, and A. Groth, *Chromatin replication and epigenetic cell memory*. Nat Cell Biol, 2020. **22**(4): p. 361-371.
27. Kelley, R.L., et al., *Epigenetic spreading of the Drosophila dosage compensation complex from roX RNA genes into flanking chromatin*. Cell, 1999. **98**(4): p. 513-22.
28. Ono, T., D. Yamashita, and T. Hirano, *Condensin II initiates sister chromatid resolution during S phase*. J Cell Biol, 2013. **200**(4): p. 429-41.
29. Sandholtz, S.H., Q. MacPherson, and A.J. Spakowitz, *Physical modeling of the heritability and maintenance of epigenetic modifications*. Proc Natl Acad Sci U S A, 2020. **117**(34): p. 20423-20429.
30. Ernst, J. and M. Kellis, *ChromHMM: automating chromatin-state discovery and characterization*. Nat Methods, 2012. **9**(3): p. 215-6.
31. Di Pierro, M., et al., *De novo prediction of human chromosome structures: Epigenetic marking patterns encode genome architecture*. Proc Natl Acad Sci U S A, 2017. **114**(46): p. 12126-12131.
32. van Steensel, B. and A.S. Belmont, *Lamina-Associated Domains: Links with Chromosome Architecture, Heterochromatin, and Gene Repression*. Cell, 2017. **169**(5): p. 780-791.
33. Wang, X. and D. Moazed, *DNA sequence-dependent epigenetic inheritance of gene silencing and histone H3K9 methylation*. Science, 2017. **356**(6333): p. 88-91.
34. Castells-Garcia, A., et al., *Super resolution microscopy reveals how elongating RNA polymerase II and nascent RNA interact with nucleosome clutches*. Nucleic Acids Res, 2022. **50**(1): p. 175-190.
35. Strom, A.R., et al., *Phase separation drives heterochromatin domain formation*. Nature, 2017. **547**(7662): p. 241-245.
36. Boija, A., et al., *Transcription Factors Activate Genes through the Phase-Separation Capacity of Their Activation Domains*. Cell, 2018. **175**(7): p. 1842-1855.e16.

37. Banigan, E.J. and L.A. Mirny, *Loop extrusion: theory meets single-molecule experiments*. *Curr Opin Cell Biol*, 2020. **64**: p. 124-138.

We sincerely thank the reviewers for their further evaluation of our manuscript and for their insightful comments and suggestions. The revisions and additional analyses conducted in response to their feedback have significantly strengthened our study. We greatly appreciate their valuable contributions to improving the manuscript.

(The reviewer comments are in plain text, our response in blue)

Reviewer #1:

The authors have adequately addressed the reviewers concerns.

Response: We thank the Reviewer for reviewing our revisions and for confirming that we have adequately addressed all concerns.

Reviewer #2:

The authors have fully addressed my comments. I recommend the paper for publication. The only optional suggestion is I find the following response paragraph from the authors very satisfyingly addressed my No. 1 question. The authors can consider including this and Fig. R2 as a supplementary analysis in the supplementary materials.

"Consistency with experimentally observed length scales: It is important to note that the interaction parameters (ϵ_{HH} and ϵ_{EE}) are tuned against experimental data such that they follow the observed spatial distance to genomic distance parameter. Hence, while tuning the A-A and B-B interaction strengths to lower values (as suggested by the reviewer) may indeed maintain chromatin domain formation and prevent them from ripening due to entropic effects, this approach will introduce discrepancies between the observed chromatin organization length scales and the computationally produced length scales. We show the effect of reducing the interaction potentials such that we have lower A-A and B-B interactions in Fig. R2 (corresponding to Fig 1d). This clearly shows that such a model would not fit the observed length scales. In contrast, the potentials in our model are calibrated to replicate experimentally observed scaling relations, ensuring consistency with available data."

Response: We sincerely thank the Reviewer for carefully evaluating our manuscript and for recommending it for publication. Following the reviewer's helpful suggestion, we have included the requested analysis (including Fig. R2, reproduced as Fig. S25, and the corresponding discussion on interaction parameter tuning and consistency with experimentally observed length scales) in the supplementary materials (section SI1.2).